# Spatiotemporally consistent global dataset of the GIMMS Leaf Area Index (GIMMS LAI4g) from 1982 to 2020

Sen Cao[1,2,3,†], Muyi Li[1,2,3,†], Zaichun Zhu[1,2,3,*], Zhe Wang[1,2,3], Junjun Zha[1,2,3], Weiqing Zhao[1,2,3], Zeyu Duanmu[1,2,3], Jiana Chen[1,2,3], Yaoyao Zheng[1,2,3], Yue Chen[1,2,3], Ranga B. Myneni[4], Shilong Piao[2,5,6]

[1]School of Urban Planning and Design, Shenzhen Graduate School, Peking University, Shenzhen 518055, China.

[2]Institute of Carbon Neutrality, Peking University, Beijing 100871, China.

[3]Key Laboratory of Earth Surface System and Human—Earth Relations, Ministry of Natural Resources of China, Shenzhen Graduate School, Peking University, Shenzhen 518055, China.

[4]Department of Earth & Environment, Boston University, Boston, MA 02215, USA.

[5]Sino-French Institute for Earth System Science, College of Urban and Environmental Sciences, Peking University, Beijing 100871, China.

[6]State Key Laboratory of Tibetan Plateau Earth System, Environment and Resources, Institute of Tibetan Plateau Research, Chinese Academy of Sciences, Beijing 100101, China.

[†]These authors contributed equally to this work

Correspondence: Zaichun Zhu (zhu.zaichun@pku.edu.cn)

**Abstract:** Leaf Area Index (LAI) with an explicit biophysical meaning is a critical variable to characterize terrestrial ecosystems. Long-term global datasets of LAI have served as fundamental data support for monitoring vegetation dynamics and exploring its interactions with other Earth components. However, current LAI products face several limitations associated with spatiotemporal consistency. In this study, we employed the Back Propagation Neural Network (BPNN) and a data consolidation method to generate a new version of the half-month 1/12° Global Inventory Modeling and Mapping Studies (GIMMS) LAI product, i.e., GIMMS LAI4g, for the period 1982−2020. The significance of the GIMMS LAI4g was the use of the latest PKU GIMMS NDVI product and 3.6 million high-quality global Landsat LAI samples to remove the effects of satellite orbital drift and sensor degradation and to develop spatiotemporally consistent BPNN models. The results showed that the GIMMS LAI4g exhibited overall higher accuracy and lower underestimation than its predecessor (GIMMS LAI3g) and two mainstream LAI products (Global LAnd Surface Satellite [GLASS] LAI and Long-term Global Mapping [GLOBMAP] LAI) using field LAI measurements and Landsat LAI samples. Its validation against Landsat LAI samples revealed an $R^2$ of 0.96, root mean squared error of 0.32 $m^2/m^2$, mean absolute error of 0.16 $m^2/m^2$, and mean absolute percentage error of 13.6% which meets the accuracy target proposed by the Global Climate Observation System. It outperformed other LAI products for most vegetation biomes in a majority area of the land. It efficiently eliminated the effects of satellite orbital drift and sensor

degradation and presented a better temporal consistency before and after the year 2000. The consolidation with the reprocessed MODIS LAI allows the GIMMS LAI4g to extend the temporal coverage from 2015 to recent (2020), producing the LAI trend that maintains high consistency before and after 2000 and aligns with the MODIS LAI trend during the MODIS era. The GIMMS LAI4g product could potentially facilitate mitigating the disagreements between studies of the long-term global vegetation changes and could also benefit the model development in Earth and environmental sciences. The GIMMS LAI4g product (Cao et al., 2023) is open access and available under Attribution 4.0 International at https://doi.org/10.5281/zenodo.7649107.

**Keywords:** GIMMS LAI4g; PKU GIMMS NDVI; Landsat LAI samples; MODIS LAI; BPNN

## 1. Introduction

Leaf Area Index (LAI), defined as half of the total green leaf area per unit of horizontal surface, is a key variable in vegetation change monitoring (Piao et al., 2015; Valderrama-Landeros et al., 2016; Zhu et al., 2016), land surface modeling (Boussetta et al., 2013; Boussetta et al., 2015; Chen et al., 2015), crop yield estimation (De Wit et al., 2012; Dente et al., 2008), etc. It is one of the basic terrestrial climate variables selected by the Global Climate Observation System (GCOS) (WMO et al., 2022). Remote sensing observation has been the only reliable means of obtaining spatiotemporally continuous LAI products at the global scale (Ma and Liang, 2022). The common practice is to relate remote sensing data with ground LAI measurements or other remote sensing products of higher reliability (as LAI reference), using methods including statistical modeling (Liu et al., 2012; Kimura et al., 2004; Broge and Leblanc, 2001), physical modeling (Myneni et al., 2002), and machine learning (Xiao et al., 2014; Zhu et al., 2013; Ma and Liang, 2022; Kang et al., 2021). Over past years, a number of long-term global LAI products, such as the third generation Global Inventory Modeling and Mapping Studies (GIMMS) LAI (GIMMS LAI3g) (Zhu et al., 2013), the Global LAnd Surface Satellite (GLASS) LAI (Xiao et al., 2016), the Long-term Global Mapping (GLOBMAP) LAI (Liu et al., 2012), and the Terrestrial Climate Data Record (TCDR) LAI (Claverie et al., 2016), have been released. These products have provided many in-depth insights into how global vegetation responds to human disturbances and global warming (Zhu et al., 2016; Piao et al., 2015; Chen et al., 2019a). Specifically, the GIMMS LAI3g has been one of the core data references in IPCC Sixth Assessment Report for the assessment of global vegetation changes (Eyring et al., 2021).

However, the accuracies of the current LAI products have been limited by uncertainties primarily in the remote sensing data and the LAI reference data (Fang et al., 2019; Jiang et al., 2017). First, remote sensing data being used have some common issues. For example, false gradual signals and mutations have been widely observed in the LAI time series prior to the late 1990s for mainstream long-term LAI products, as most of them utilized data from the Advanced Very High Resolution Radiometer (AVHRR) (Wang et al., 2022). The AVHRR sensors on board National Oceanic and Atmospheric Administration

(NOAA) satellites were the only remote sensing data sources before the late 1990s that provided spatiotemporally continuous observations over the globe. Nevertheless, they suffered from issues of NOAA satellite orbital drift and AVHRR sensor degradation, particularly in the tropical area of evergreen broadleaf forests (Pinzon and Tucker, 2014). Second, the LAI reference data used to build LAI models have been scarce, particularly before the late 1990s. After the year of 2000, global LAI products became increasingly available from advanced sensors such as the Moderate Resolution Imaging Spectroradiometer (MODIS) (Myneni et al., 2002), the Système Pour l'Observation de la Terre (SPOT) (Baret et al., 2007), and the Visible Infrared Imaging Radiometer Suite (VIIRS) (Yan et al., 2018). These LAI products have been elaborately and collaboratively validated despite with a short time span. Current studies employed them as the LAI reference to build post-2000 AVHRR−LAI models and extrapolated the models onto pre-2000 AVHRR data so that long-term LAI products could be produced (Chen et al., 2019a). Nonetheless, the legality of the extrapolation remains questioned since the AVHRR−LAI relationship could change with time.

The uncertainties in the remote sensing and LAI reference data, together with the differences in modeling algorithms, have led the performance of long-term global LAI products to vary from one to another. Inconsistencies were continually found between LAI products regardless of the remote sensing data source used (Jiang et al., 2017). For example, four popular global data sets of LAI (1982−2010s), namely the GLASS LAI, GLOBMAP LAI, GIMMS LAI3g, and TCDR LAI, showed significant differences in LAI trends, interannual variabilities, and uncertainty variations (Jiang et al., 2017; Xiao et al., 2017). In tropical areas, the average LAI difference can be up to one unit (Yan et al., 2016). These differences between LAI products have raised many concerns about the robustness of existing vegetation change analysis and land surface modeling (Alkama et al., 2022; Jiang et al., 2017; Piao et al., 2015).

Recent advances in land data products have provided pathways to address the uncertainties. In particular, the PKU GIMMS Normalized Difference Vegetation Index (NDVI) product (1982−2022) by Li et al. (2023a) efficiently eliminated the evident NOAA orbital drift and AVHRR sensor degradation effects. It demonstrates higher accuracy than the predecessor (GIMMS NDVI3g) and shows a high temporal consistency with MODIS NDVI. Zha et al. (2023) compiled a set of global reference LAI (before the year 2000) and created ~ 4.9 million high-quality Landsat LAI samples over the globe from 1984 to 2020. The validation against LAI field measurements showed an $R^2$ of 0.76. Although these Landsat LAI samples can hardly be used to characterize the global vegetation change because they are not spatiotemporally continuous, they can serve as reliable LAI references.

In this context, the objective of this study is to derive a new generation of GIMMS LAI products (GIMMS LAI4g, 1982−2020) using machine learning models based on the PKU GIMMS NDVI product and massive high-quality Landsat LAI samples and a data consolidation method based on the Reprocessed MODIS LAI product. We employ the PKU GIMMS NDVI and the Landsat LAI samples to address the uncertainties in remote sensing and LAI reference data. With these data, biome-specific Back Propagation Neural Network (BPNN) models are developed with additional explanatory variables (the longitude and latitude, the NDVI month, and the NOAA number and years since launch). The GIMMS LAI4g product is then generated

from the BPNN models. Finally, the GIMMS LAI4g is consolidated with the Reprocessed MODIS NDVI product via a pixel-wise fusion method to extend the temporal coverage to the year 2020. We evaluate the GIMMS LAI4g's accuracy by a direct validation method and compare its accuracy to those of three other global LAI products, i.e., GIMMS LAI3g, GLASS LAI, and GLOBMAP LAI. The temporal consistency of the global LAI products and their LAI trends are also analyzed.

## 2. Data

A total of eight global datasets were used in this study, namely, the PKU GIMMS NDVI, Landsat LAI sample dataset, MODIS Land-Cover Type, Reprocessed MODIS LAI, GLASS LAI, GLOBMAP LAI, GIMMS LAI3g, and field LAI measurements. The PKU GIMMS NDVI was the primary data source from which the GIMMS LAI4g was generated. The Landsat LAI sample dataset was used as the LAI reference in machine learning model establishment and product evaluation. The field LAI measurements were also employed for product evaluation. The MODIS Land-Cover Type product provided vegetation biome types in the LAI modeling. The Reprocessed MODIS LAI was used to extend the temporal coverage of the GIMMS LAI4g. The GLASS LAI, GLOBMAP LAI, and GIMMS LAI3g are three mainstream global LAI products that were included for an inter-comparison purpose.

### 2.1 PKU GIMMS NDVI

The PKU version of the GIMMS NDVI product (PKU GIMMS NDVI) was employed in this study (Li et al., 2023a; Li et al., 2023b). It has a spatial resolution of 1/12° and a temporal resolution of half-month. In the generation of PKU GIMMS NDVI, Landsat NDVI from Thematic Mapper (TM), Enhanced Thematic Mapper Plus (ETM+), and Operational Land Imager (OLI) were first cross-calibrated by adjusting the TM and OLI NDVI to the ETM+ level via random sample locations and the BPNN model (Berner et al., 2020). The sample locations were refined by removing those with high atmospheric opacity and low quality which was defined by the occurrence of clouds, cloud shadows, water, or snow and implausible radiation performance. In the BPNN model, the explanatory variables included the NDVI of TM or OLI, the image acquisition day of the year, and the sample location's longitude and latitude; and the target variable was the NDVI of ETM+.

After cross-calibration, massive high-quality Landsat NDVI samples were extracted by screening out samples that suffered from the Mount Pinatubo eruption (August 1991 to December 1992) as well a high atmospheric opacity and a bad quality (same as sample screening in cross-calibration). The Landsat NDVI samples were employed to calibrate the GIMMS NDVI3g product with other explanatory variables (the longitude and latitude, NDVI month, and NOAA satellite number and years since its launch) using biome-specific machine learning models. The calibrated NDVI product was finally consolidated with the MODIS NDVI product to extend the temporal coverage to the year 2020.

The major improvement of PKU GIMMS NDVI over its counterparts is that it well removed the NOAA orbital drift and AVHRR sensor degradation effects, especially in tropical regions (Figure S1). Its overall $R^2$, Mean Absolute Error (MAE), and Mean Absolute Percentage Error (MAPE) are 0.975, 0.033, and 9%, respectively. It is highly consistent with MODIS NDVI in terms of pixel value ($R^2$ = 0.962, MAE = 0.032, and MAPE = 6.5%) and global vegetation trend. PKU GIMMS NDVI inherited the quality control (QC) information from the GIMMS NDVI3g. A QC value of 0, 1, and 2 indicates NDVI of good quality, NDVI retrieved from spline interpolation, and NDVI retrieved from average seasonal profile, respectively. The PKU GIMMS NDVI record during AVHRR missions from 1982 to 2015 (before consolidation with MODIS NDVI) was used in this study. It is available at https://doi.org/10.5281/zenodo.7441558 (last access: September 2023).

## 2.2 Landsat LAI sample dataset

The Landsat LAI sample dataset provides approximately 4.9 million high-quality samples with a spatial resolution of 1/12 ° and a temporal resolution of half a month (Zha et al., 2023). It covers the global vegetated area with all vegetation biome types defined in the MODIS land cover product (the third classification scheme; see section 2.4) and a long-time span from 1984 to 2020. In the generation of Landsat LAI samples, 70,000 sample locations for Deciduous Needleleaf Forests [DNF] and 100,000 sample locations for each of the other vegetation biome types were randomly selected based on the MODIS land cover product. At the sample locations, Reprocessed MODIS LAI (in 500 m resolution; see Section 2.3) and Landsat surface reflectance from TM, ETM+, and OLI scenes (20 × 20 pixels in 30 m resolution) were extracted, creating massive sample pairs. The sample pairs were then rigorously screened by criteria that were not limited to those mentioned in Section 2.1 (i.e., clouds, cloud shadows, etc.) but also included Landsat sample purity, NDVI-LAI relationship, and the saturation state of the MODIS LAI. Biome- and Landsat sensor-specific Random Forest models with other explanatory variables (NDVI, Normalized Difference Water Index [NDWI], Enhanced Vegetation Index [EVI], the longitude and latitude, and the solar zenith and azimuth angles) were built based on the sample pairs. The models were applied to historical Landsat data at 40,000 random sample locations (1/12°) to create the final Landsat LAI sample dataset. Validation of the dataset through observations from the BEnchmark Land Multisite ANalysis and Intercomparison of Products (BELMANIP) network (Baret et al., 2006) and the Oak Ridge National Laboratory (ORNL) (Scurlock et al., 2001) showed high absolute accuracies ($R^2$ = 0.76, MAE = 0.45 $m^2/m^2$, Root Mean Square Error (RMSE) = 0.66 $m^2/m^2$). The inter-comparison with the Reprocessed MODIS LAI shows a high temporal consistency. This study selected 3.6 million Landsat LAI samples between 1984 and 2015.

## 2.3 Reprocessed MODIS LAI product

The latest version of the Reprocessed MODIS LAI product (version 6) has a time span of 2000−2020, a temporal resolution of 8-day or one-month, and a spatial resolution ranging from 500 m to 0.5°. The product was derived from the MODIS LAI Version 6 products (Myneni, 2015a, b) and MODIS Land Cover Type product (Friedl and Sulla-Menashe, 2022)

using an integrated two-step method (Yuan et al., 2011). Compared to the original MODIS LAI products, it is more spatiotemporally continuous and consistent as verified by 44 LAI reference maps which contain true LAI values collected over a subset of 26 ground sites. We downloaded the 8-day 0.05° data from http://globalchange.bnu.edu.cn/research/laiv6#download (last access: September 2023), and resampled the data to have the
155 same spatial resolution (1/12°) and temporal resolution (half a month) as the GIMMS LAI4g. The temporal subset of 2004−2020 was used in this study because the LAI data in the evergreen broadleaf forest were found exceptionally low between 2000 and 2003 than other years (Figure S2).

## 2.4 MODIS Land-Cover Type product (MCD12Q1)

The MODIS Land Cover Type Product (MCD12Q1, version 6.1) supplies global maps of annual land cover with a
160 spatial resolution of 500 m since 2001 (Friedl et al., 2002; Friedl and Sulla-Menashe, 2022). It includes five legacy classification schemes. This study selected the third classification scheme (Annual LAI classification). The Annual LAI classification scheme includes eight natural vegetation types (Evergreen Needleleaf Forests [ENF], Evergreen Broadleaf Forests [EBF], DNF, Deciduous Broadleaf Forests [DBF], SHRublands [SHR], SAVannas [SAV], GRAsslands [GRA], CROplands [CRO]), and three non-vegetated lands (WATer bodies [WAT], Non-VeGetated lands [NVG], and URban and
165 Built-up lands [URB]). This study also used [GLO] in data analysis to represent the global vegetation biome (the ensemble of the eight vegetation types). The spatial resolution of MCD12Q1 was spatially aggregated to 1/12° in this study to match that of PKU GIMMS NDVI. For each 1/12° grid, the aggregation was conducted by calculating frequencies of each biome type between 2001 and 2019 and identifying the most frequent one. This generated a global land cover map that was considered static from 1982 through 2020 in this study (Figure S3). With potential errors, this strategy could be the best option at the time.

## 2.5 GLASS LAI

The GLASS LAI (version 4) with a temporal resolution of 8 days was generated from the 0.05° resolution NOAA/AVHRR surface reflectance dataset provided by NASA's Long Term Data Record (LTDR) project (1982−2000) and the 1 km resolution Terra/MODIS surface reflectance dataset (MOD09) (2000−2018) (Xiao et al., 2016). In the algorithm, biome-specific general regression neural networks were built between the surface reflectance data and LAI reference data which were created by fusing Terra/MODIS LAI (MOD15) with clump-corrected CYCLOPES LAI over BELMANIP sites
(Xiao et al., 2016). The neural networks were then used to predict global LAI (Xiao et al., 2014). The GLASS LAI (V4) product was acquired from ftp://ftp.glcf.umd.edu/. It should be noted that version 5 and version 6 of the GLASS LAI product have been available when our study has been prepared (Liang et al., 2021; Ma and Liang, 2022).

### 2.6 GLOBMAP LAI

The latest GLOBMAP LAI product (version 3) with a spatial resolution of 1/13.75° and temporal resolutions of half-month (1982−2000) or 8-day (2001−now) was generated based on GIMMS NDVI product (1982−2000) (Tucker et al., 2005) and Terra/MODIS surface reflectance (MOD09A1 C6) (2001−now). The algorithm established relationships between MODIS LAI and GIMMS NDVI in a pixel-wise manner during their overlapping period of 2000−2006. The relationships were then applied to GIMMS NDVI before 2000 (Liu et al., 2012). The GLOBMAP LAI (V3) product was acquired from https://doi.org/10.5281/zenodo.4700264 (last access: September 2023) (Liu et al., 2021).

### 2.7 GIMMS LAI3g

The GIMMS LAI3g product (version 4) (1982−2016) was generated biweekly in a 1/12° spatial resolution (Zhu et al., 2013). It is available at https://drive.google.com/drive/folders/0BwL88nwumpqYaFJmR2poS0d1ZDQ?resourcekey=0-9IRE9s-0tFGfwB5qTpLjZw&usp=sharing/ (last access: September 2023). The algorithm related the GIMMS' third-generation NDVI (NDVI3g) to the Reprocessed MODIS LAI product via feed-forward neural networks (Yuan et al., 2011). Twelve neural networks, one for each month, were built using monthly averaged LAI and NDVI data between 2000 and 2009. The GIMMS LAI3g was then produced from GIMMS NDVI3g by applying the neural networks to the period of 1982 to 2016.

### 2.8 Field LAI measurements

The field LAI measurements were from three projects namely, BELMANIP 2.1 (available at https://calvalportal.ceos.org/web/olive/site-description, last access: September 2023) (Baret et al., 2006), DIRECT 2.1 (available at https://calvalportal.ceos.org/lpv-direct-v2.1, last access: September 2023) (Garrigues et al., 2008), and ORNL (available at https://daac.ornl.gov/VEGETATION/guides/LAI_guide.html, last access: September 2023) (Scurlock et al., 2001). The BELMANIP 2.1 and DIRECT 2.1 provide 3 km × 3 km averaged LAI values derived from sites in networks of FLUXNET, AERONET, VALERI, BigFoot, etc. The upscaling from site-based LAI to 3 km × 3 km LAI used high spatial resolution imageries such as Landsat and SPOT. Most global long-term LAI products have utilized the BELMANIP and DIRECT LAI as ground truth for product evaluation (Myneni et al., 2002; Liu et al., 2012; Xiao et al., 2016; Zhu et al., 2013), yet the LAI measurements in both projects were available only after the late 1990s. Note that GLASS LAI (version 4) also employed BELMANIP sites for LAI model training (Xiao et al., 2016). This study further incorporated ORNL sites which provided field LAI measurements during 1932−2020 despite possible scaling effects due to spatial heterogeneity. We prudently examined all the measurements in BELMANIP 2.1, DIRECT 2.1, and ORNL, and removed those that were acquired from heterogeneous sites using an 8 km × 8 km window (approximately 1/12°). Redundant measurements among the three projects were also removed. In a spatial resolution of 1/12° and a temporal resolution of half-moth, we averaged the measurements falling in the

same spatial or temporal domain. Eventually, 113 field LAI measurements from 49 sites were obtained. Information on selected field LAI measurements can be found in Table S1.

## 3. Methodology

The methodology includes three key steps (Figure 1): 1) generating the GIMMS LAI4g product from biome-specific BPNN models based on PKU GIMMS NDVI, Landsat LAI sample, and other explanatory variables; 2) consolidating the GIMMS LAI4g product with the MODIS LAI product using a pixel-wise fusion method in their overlapping timespan (2004−2015); and 3) evaluating the GIMMS LAI4g product using Landsat LAI samples and comparing it with other global LAI products.

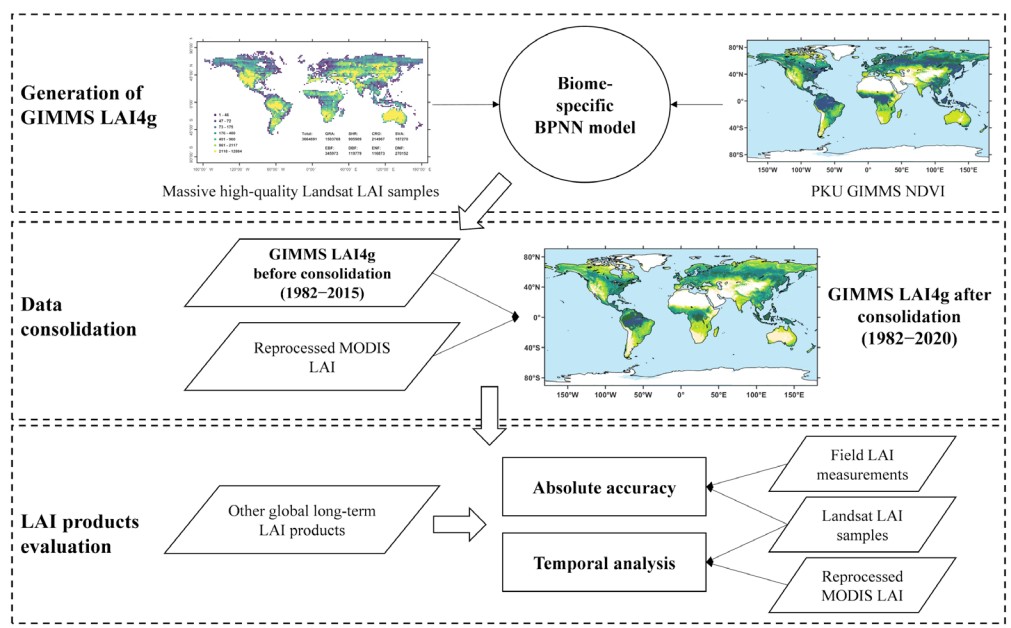

**Figure 1.** Schematic diagram of the generation and evaluation of the GIMMS LAI4g product.

### 3.1 Generation of GIMMS LAI4g using Back Propagation Neural Network (BPNN)

The artificial neural network (ANN) is a machine-learning algorithm inspired by the structure and function of biological neural networks (Basheer and Hajmeer, 2000; Zhang et al., 1998). It has been frequently used in ecological studies and the generation of global LAI products (Panda et al., 2010; Jahan and Gan, 2011; Zhu et al., 2013; Xiao et al., 2014; Claverie et al., 2016). For example, a typical ANN, general regression neural network, and BPNN was employed in the production of TCDR

LAI (version 5) (Claverie et al., 2016), GLASS LAI (version 4) (Xiao et al., 2014), and GIMMS LAI3g (Zhu et al., 2013),
respectively. A typical ANN comprises input, output, and hidden layers, with each containing several artificial neurons. During the model training process, signals flowed from the input layer to the output layer, after likely passing through several hidden layers. Errors in the output layer propagate backward to the previous layers until they satisfy the user-defined threshold, and the network attempts to minimize the discrepancies between observations and predictions (Basheer and Hajmeer, 2000; Zhang et al., 1998).

This study used the BPNN model to predict LAI values from the PKU GIMMS NDVI (1982−2015). Individual BPNN models were developed for each vegetation biome. The target variable in BPNN models was mainly from the Landsat LAI samples (1984−2015) but also included 40,000 MODIS LAI values in regions and months where Landsat LAI samples were lacking. These regions were mostly located in northern high latitudes that suffer from polar night phenomena and low solar altitude angles in the winter. Specifically, 10,000 Reprocessed MODIS LAI values were randomly introduced for each of GRA,
SHR, SAV, and ENF at latitudes > 25° N in the winter months (October to April). Corresponding PKU GIMMS NDVI values of the same time and at the same locations with the LAI samples were extracted as the explanatory variable. The LAI samples and associated PKU GIMMS NDVI were further refined. Locations with negative NDVI values (e.g., contaminated by snow and inland water bodies) and non-zero QC values in the PKU GIMMS NDVI product were removed. After the refinement, the samples were randomly divided into two groups, i.e., the dataset for BPNN construction (80%) and the dataset for LAI product
evaluation (20%).

In the BPNN models, we also incorporated spatial information (the longitude and latitude at the sample location), temporal information (the NDVI month), and NOAA satellite information (NOAA satellite number and years since launch; not applicable for MODIS LAI) as additional explanatory variables. A stepwise method was employed to determine the best combination of explanatory variables for each vegetation biome. The PKU GIMMS NDVI data were first included and
evaluated in the BPNN models (Scenario 1 or S1). Then, the spatial information that accounts for spatial autocorrelation (S2), temporal information that accounts for vegetation dynamic (S3), NOAA satellite number (S4), and years since NOAA launch (S5) that account for potential satellite and sensor issues were added one by one. In model establishment, we repeatedly (50 times) selected 50,000 random samples with replacement for each vegetation biome. The 50,000 samples were split into 90% for model training and 10% for model evaluation, in which four error metrics of $R^2$, RMSE ($m^2/m^2$), MAE ($m^2/m^2$), and MAPE
(%), were calculated. The error metrics determined the optimum combination of explanatory variables and the optimum parameters for the final BNPP model of each biome.

## 3.2 Consolidation of GIMMS LAI4g and MODIS LAI

The GIMMS LAI4g product derived from the PKU GIMMS NDVI (1982−2015) which was based on AVHRR data did not include LAI data since 2015. As such, it can hardly be used to characterize recent vegetation dynamics. A couple of global
products have provided up-to-date LAI data using satellite sensors available since the late 1990s (Baret et al., 2007). A common

practice to generate the long-term LAI product is to consolidate the AVHRR-based LAI product with the post-2000 LAI product. For example, both GLASS LAI and GLOBMAP LAI consolidated LAI products from AVHRR and MODIS. MODIS has been one of the most popular and verified data sources for LAI production. In this study, the Reprocessed MODIS LAI product (2004−2020) was employed to extend the time span of the GIMMS LAI4g.

The consolidation method was inherited from the pixel-wise linear fusion method proposed by Mao et al. (2012). Compared to the global or biome-specific regression models, the pixel-wise method has demonstrated excellent accuracies, especially in regulating the temporal consistency between datasets (Mao et al., 2012). First, LAI values in the overlapping period of 2004−2015 were extracted from the GIMMS LAI4g and the MODIS LAI. Then, the most appropriate Random Forest regression model (Breiman, 2001) (instead of the linear model) was determined from the LAI values at an 11 × 11 window

(approximately 1° equivalent) around each pixel location, with GIMMS LAI4g data and the pixel coordinates as the explanatory variables and MODIS LAI data as the target variable. The final LAI product comprised the GIMMS LAI4g (after consolidation) (1982−2003) and the Reprocessed MODIS LAI (2004−2020).

## 3.3 Evaluation of the GIMMS LAI4g product

In this study, the LAI reference samples were evaluated in terms of their number, spatial distribution, and temporal

distribution under different vegetation biome types. To assess the representativeness of the samples, we also compared LAI reference values at the sample locations to those from the GIMMS LAI3g, GLASS LAI, and GLOBMAP LAI using a frequency histogram.

The performance of our GIMMS LAI4g product generated from the BPNN model was evaluated and compared with three other global long-term LAI products (i.e., the GIMMS LAI3g, GLASS LAI, and GLOBMAP LAI) using field LAI

measurements and Landsat LAI samples. Four measures of error were used: $R^2$, RMSE ($m^2/m^2$), MAE ($m^2/m^2$), and MAPE (%). $R^2$ measures the percentage of variations that models can explain; RMSE quantifies the variance of errors; and MAE and MAPE measure absolute and relative error values at the sample level. For inter-comparison between the GIMMS LAI4g and other LAI products, the spatial resolution and temporal resolution of all LAI products have been unified to 1/12° and half a month, respectively. In the validation against Landsat LAI samples, the remaining 20% Landsat sample points with a spatial

resolution of 1/12° were employed for each vegetation biome. Based on the Landsat sample points, we used a dominance map to demonstrate the global distribution of products with higher accuracy. The map was drawn with 2° × 2° grids using MAE values from the GIMMS LAI4g, GIMMS LAI3g, and GLASS LAI. The color of each grid was composed of reciprocal MAE averages of the LAI products, i.e., a lower MAE average can lead to a higher weight in the composite. The GLOBMAP LAI was not included because of its much higher MAE than other products. We also showcased the spatial consistencies between

the four global LAI products by their spatial average along latitude (at an interval of 1°) in January and July of the years 1990, 2000, and 2010.

The temporal consistency of the GIMMS LAI4g was evaluated from three perspectives. First, LAI bias was used to examine whether the NOAA orbital drift and AVHRR sensor degradation effects were alleviated in different vegetation biomes. The bias was calculated as the mean value of LAI deviation relative to Landsat LAI in percentage (Helder et al., 2013). If there is orbital drift or sensor degradation, the bias will drastically fluctuate; otherwise, it remains constant. Seasonal fluctuations in the time series of LAI bias were first removed via the multi-year averaging method. Then, inter-annual trends of the bias were extracted via the Ensemble Empirical Mode Decomposition (EEMD) approach (Huang et al., 1998). Second, the efficiency of data consolidation between the GIMMS LAI4g and MODIS LAI was reported. We also checked the self-consistency of the GIMMS LAI4g product over time in some hotspot regions including Europe (Ciais et al., 2005), Amazon (Wang et al., 2013), Congo (Zhou et al., 2014), China, and India (Chen et al., 2019a). Third, we used the Landsat LAI samples as the reference to evaluate the consistency of the GIMMS NDVI4g between different periods (p1: 1984−2015; p2: 1984−2000; and p3: 2001−2015) and compared the consistency with the other three LAI products. The consistency was quantified by temporal changes in $R^2$, MAE, and MAPE. To investigate whether the data consolidation alter the LAI trend, we compared the annual anomalies and trends of GIMMS LAI4g before consolidation (1982−2015), GIMMS LAI4g after consolidation (1982−2020), Reprocessed MODIS LAI (2004−2020), and PKU GIMMS NDVI (1982−2015).

The LAI trends between 1982 and 2015 were derived and compared between the four global LAI products. Linear regression analysis was performed on the LAI time series at the pixel level. The trend was calculated as the slope of the fitting line, which indicates greening (positive slope) or browning (negative slope). This produced a global map of LAI trending. We also analyzed annual LAI variations during 1982−2015 and paid special attention to vegetation trends before and after 2000 for all vegetation biome types. The annual LAI value is an area-weighted average based on the vegetation biome type.

## 4. Results

### 4.1 Examination of LAI reference data

The spatial and temporal distributions of the LAI reference data were determined mostly by the availability of Landsat images but also by the occurrence of cloud cover, aerosol, and other factors. Figure 2a shows the spatial distribution of the 3.6 million LAI samples primarily from the Landsat LAI dataset. The sample size for each vegetation biome, ranging from 116,873 (ENF) to 1,503,768 (GRA) is also listed. The sample locations spanned all latitudes of the vegetated area, and no samples were selected from non-vegetation regions. The sample size per biome was approximately proportionate to the biome area (Figure 2b). In northern high latitudes, Landsat images were scarce throughout the winter due to the polar night phenomenon and the low solar altitude angle; and in the tropical area, Landsat images were frequently contaminated by precipitation and clouds. As a result, the number of available samples was limited in these two areas (Figure 2a). We addressed this issue by introducing 40,000 samples from the Reprocessed MODIS LAI at locations and months when Landsat LAI samples were scarce. During

1984−2015, the Landsat LAI sample size per year increased from 28,323 in 1984, peaked at 200,315 in 2001 when both Landsat 5 and Landsat 7 were available, and then leveled off until 2012 (sample size: 22,106) (Figure 2b). From November 2011 to May 2012, very few images were acquired with Landsat 5's decommissioning, (https://www.usgs.gov/centers/eros/science/usgs-eros-archive-landsat-archives-landsat-4-5-thematic-mapper-tm-level-1-data, last access: September 2023). Since the launch of Landsat 8 in 2013, the Landsat LAI sample became steadily available again.

To evaluate the representativeness of the Landsat LAI samples, we calculated a frequency histogram based on all Landsat LAI sample values between 1984 and 2015 and compared it to those based on GIMMS LAI3g, GLASS LAI, and GLOBMAP LAI (Figure 2c). During 1984−2015, the LAI value distribution in the Landsat samples was similar to those in the other three products at global vegetation pixels (Figure 2c), indicating that the Landsat LAI samples used in this study have good representativeness.

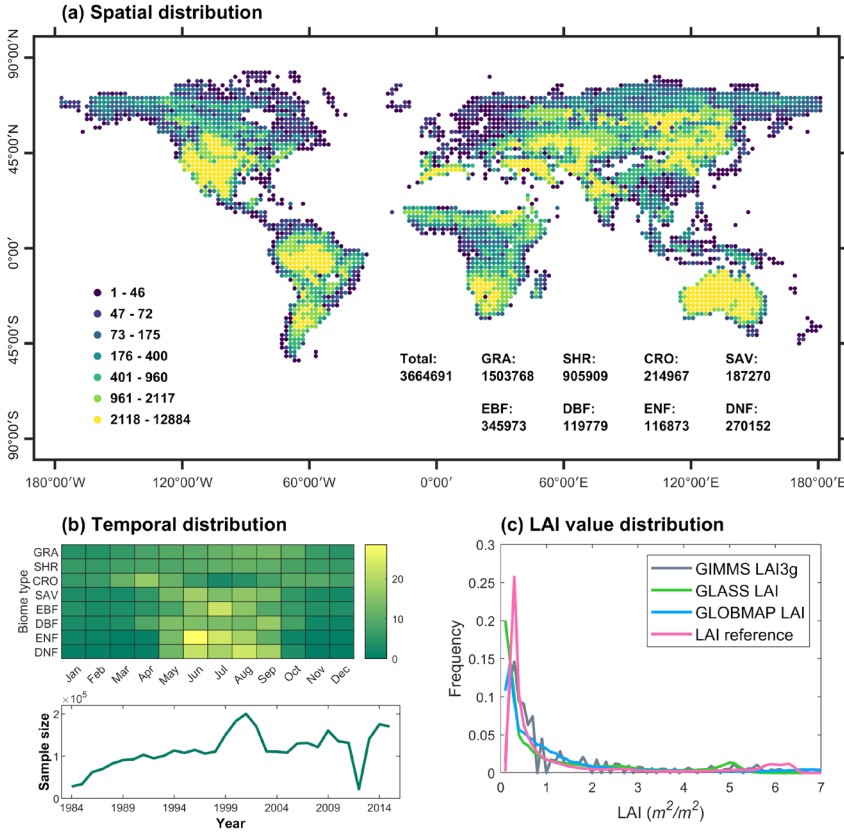

**Figure 2.** Spatial, temporal, and value distribution of the LAI reference data. (a) The global distribution of LAI samples in 2° grids. The LAI sample size for each vegetation is listed. (b) The temporal distribution of LAI samples for the eight vegetation biome types and the annual variation of LAI sample size. (c) The distribution of LAI values in percentage (bin width = 0.1) for Landsat LAI samples, GIMMS LAI3g, GLASS LAI, and GLOBMAP LAI. For GIMMS LAI3g, GLASS LAI, and GLOBMAP LAI, the value distribution was calculated based on

all terrestrial vegetation pixels. It should be noted that 40,000 Reprocessed MODIS LAI samples were introduced at locations and months when Landsat LAI samples were scarce.

## 4.2 The optimum BPNN models

335  For each vegetation biome, different combinations of explanatory variables (S1 to S5, see section 3.1) were tested in BNPP models. The variations in accuracy are shown in Figure 3 and Table 1. The inclusion of spatial information and temporal information has significantly improved the model performance with much higher $R^2$, lower RMSE, lower MAE, and lower MAPE for most vegetation biomes (Figure 3). The improvement from spatial information was slight for DNF, probably because of its relative concentration in certain middle- and high-latitudes of Eurasia (Figure S3). The inclusion of NOAA satellite

340  number and years since its launch brought subtle but discernible improvements towards the accuracy of BPNN models.

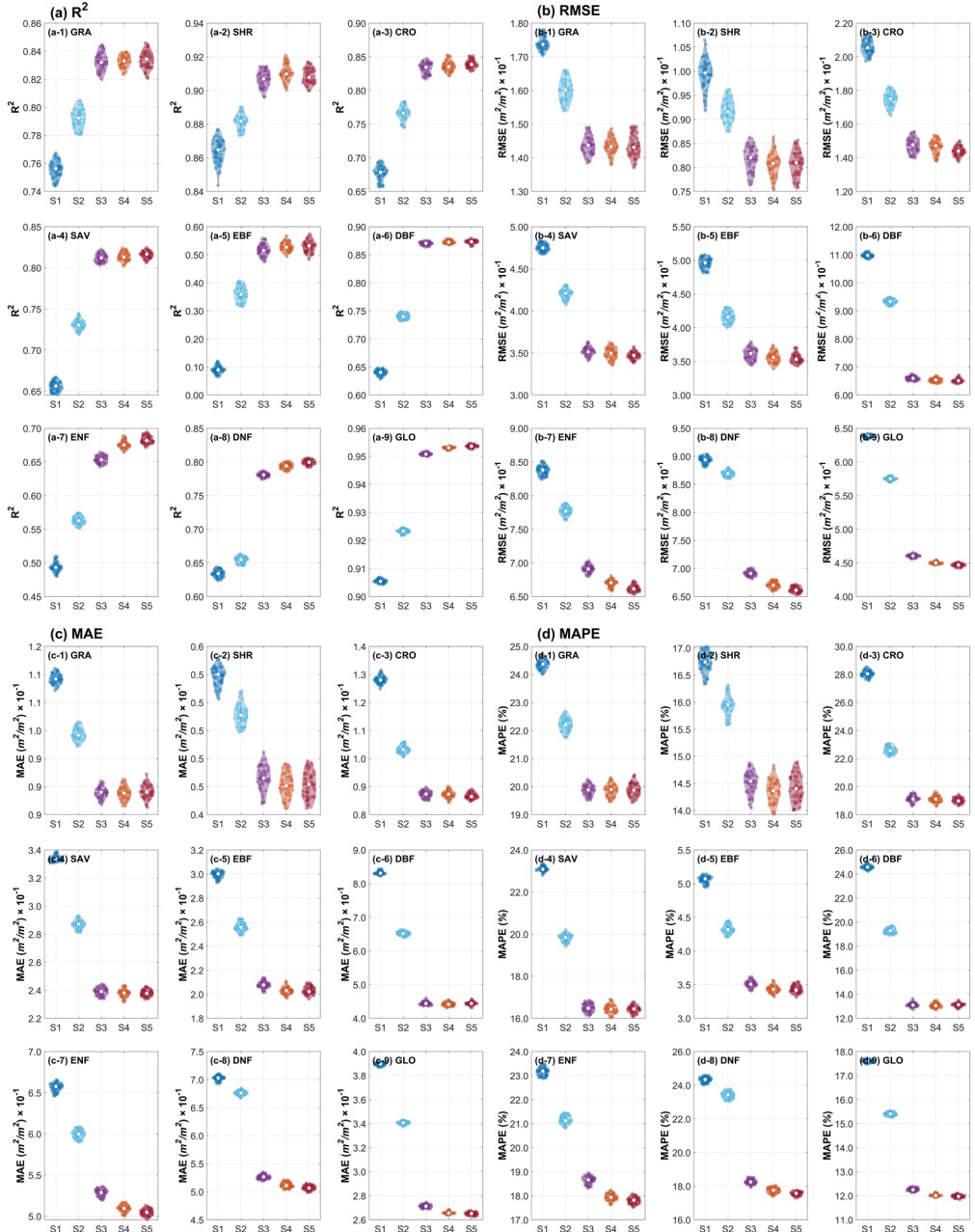

**Figure 3.** Performance of different combinations of explanatory variables (S1 to S5) in BPNN models for each vegetation biome. (a), (b), (c), and (d) shows the R², RMSE, MAE, and MAPE, respectively, calculated based on Landsat LAI samples. GLO represents the global vegetation biome. The combinations of explanatory variables are (S1) NDVI alone; (S2) NDVI and spatial information (longitude and latitude); (S3) NDVI, spatial information, and temporal information (month); (S4) NDVI, spatial information, temporal information, and NOAA satellite number; and (S5) NDVI, spatial information, temporal information, NOAA satellite number and years since its launch.

Using all explanatory variables in the BPNN model (S5) has resulted in $R^2$ values > 0.80 for most biome types except EBF (0.53) and ENF (0.68), RMSE values < 0.66 $m^2/m^2$, MAE < 0.51 $m^2/m^2$ and MAPE values < 20% (Table 1). MAPE of EBF was as low as 3.43%. For the global vegetation biome as a whole, accuracies of the BPNN model in S5 were $R^2$ of 0.95, RMSE of 0.45 $m^2/m^2$, MAE of 0.27 $m^2/m^2$, and MAPE of 11.98% (Table 1). As such, for most periods during 1982−2015, the BPNN models adopted the combination of all explanatory variables (S5), including NDVI, longitude, latitude, month, NOAA satellite number, and NOAA satellite in orbit duration. For the period of 1982−1984, the BPNN models adopted the combination of NDVI, longitude, latitude, and month (S3) because of the acceptable accuracies (Figure 3 and Table 1) and the absence (before 1984) and scarce (1984) Landsat LAI samples (see section 4.1) which could lead to a biased derivation of LAI. Similarly, S3 was also adopted in the winter of ENF (Northern Hemisphere: October to April; Southern Hemisphere: May to September) and October to April of EBF due to the limited Landsat LAI samples (Figure 2).

Table 1 Error metric values for different combinations of explanatory variables (S1 to S5) in BPNN of each vegetation biome. Values in this table correspond to Figure 3. The combinations of explanatory variables are (S1) NDVI alone; (S2) NDVI and spatial information (longitude and latitude); (S3) NDVI, spatial information, and temporal information (month); (S4) NDVI, spatial information, temporal information, and NOAA satellite number; and (S5) NDVI, spatial information, temporal information, NOAA satellite number and years since its launch. GLO represents the global vegetation biome.

| Metrics | combinations | Biome type | | | | | | | | |
|---|---|---|---|---|---|---|---|---|---|---|
| | | GRA | SHR | CRO | SAV | EBF | DBF | ENF | DNF | GLO |
| $R^2$ | S1 | 0.76 | 0.86 | 0.68 | 0.66 | 0.09 | 0.64 | 0.49 | 0.63 | 0.91 |
| | S2 | 0.79 | 0.88 | 0.77 | 0.73 | 0.36 | 0.74 | 0.56 | 0.65 | 0.92 |
| | S3 | 0.83 | 0.91 | 0.83 | 0.81 | 0.52 | 0.87 | 0.65 | 0.78 | 0.95 |
| | S4 | 0.83 | 0.91 | 0.84 | 0.81 | 0.53 | 0.87 | 0.68 | 0.79 | 0.95 |
| | S5 | 0.83 | 0.91 | 0.84 | 0.82 | 0.53 | 0.87 | 0.68 | 0.80 | 0.95 |
| RMSE ($m^2/m^2$) | S1 | 0.17 | 0.10 | 0.21 | 0.48 | 0.50 | 1.10 | 0.84 | 0.89 | 0.64 |
| | S2 | 0.16 | 0.09 | 0.17 | 0.42 | 0.42 | 0.93 | 0.78 | 0.87 | 0.57 |
| | S3 | 0.14 | 0.08 | 0.15 | 0.35 | 0.36 | 0.66 | 0.69 | 0.69 | 0.46 |
| | S4 | 0.14 | 0.08 | 0.15 | 0.35 | 0.36 | 0.65 | 0.67 | 0.67 | 0.45 |
| | S5 | 0.14 | 0.08 | 0.14 | 0.35 | 0.36 | 0.65 | 0.66 | 0.66 | 0.45 |
| MAE ($m^2/m^2$) | S1 | 0.11 | 0.05 | 0.13 | 0.33 | 0.30 | 0.83 | 0.66 | 0.70 | 0.39 |
| | S2 | 0.10 | 0.05 | 0.10 | 0.29 | 0.26 | 0.65 | 0.60 | 0.68 | 0.34 |
| | S3 | 0.09 | 0.05 | 0.09 | 0.24 | 0.21 | 0.45 | 0.53 | 0.53 | 0.27 |
| | S4 | 0.09 | 0.05 | 0.09 | 0.24 | 0.20 | 0.44 | 0.51 | 0.51 | 0.27 |
| | S5 | 0.09 | 0.05 | 0.09 | 0.24 | 0.20 | 0.45 | 0.50 | 0.51 | 0.27 |
| MAPE (%) | S1 | 24.36 | 16.75 | 28.06 | 23.09 | 5.06 | 24.60 | 23.16 | 24.33 | 17.61 |
| | S2 | 22.18 | 15.95 | 22.59 | 19.82 | 4.33 | 19.29 | 21.16 | 23.40 | 15.40 |
| | S3 | 19.85 | 14.55 | 19.17 | 16.50 | 3.51 | 13.15 | 18.64 | 18.26 | 12.27 |
| | S4 | 19.88 | 14.36 | 19.11 | 16.45 | 3.44 | 13.09 | 17.94 | 17.73 | 12.02 |
| | S5 | 19.88 | 14.45 | 19.03 | 16.46 | 3.43 | 13.16 | 17.79 | 17.56 | 11.98 |

**4.3 Validation of the GIMMS LAI4g and other LAI products**

Based on field LAI measurements, GIMMS LAI4g generated from the BPNN models presented comparable accuracies ($R^2 = 0.70$, RMSE = 0.86 $m^2/m^2$, MAE = 0.60 $m^2/m^2$, MAPE = 32.8%) with GIMMS LAI3g ($R^2 = 0.72$, RMSE = 0.78 $m^2/m^2$, MAE = 0.56 $m^2/m^2$, MAPE = 30.4%) and GLASS LAI ($R^2 = 0.68$, RMSE = 0.83 $m^2/m^2$, MAE = 0.60 $m^2/m^2$, MAPE = 32.8%) (Figure 4). GIMMS LAI3g had the best performance in error measures (i.e., $R^2$, RMSE, MAE, and MAPE), but GIMMS LAI4g had the lowest underestimation for the fitting line with a slope of 0.90 and an intercept of 0.03 (Figure 4). GLOBMAP

LAI presented the largest discrepancies from the LAI measurements.

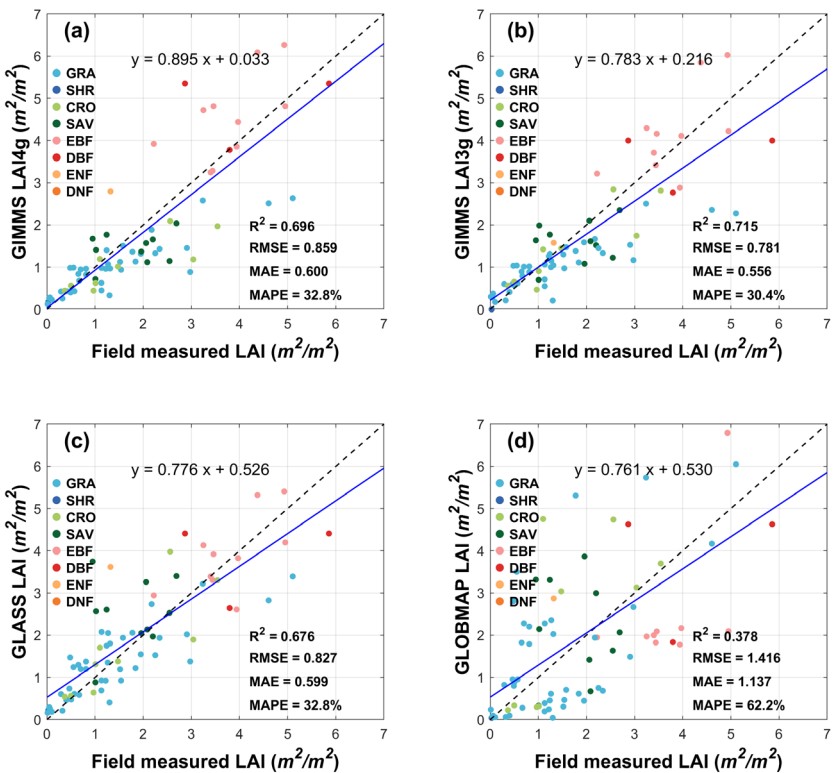

**Figure 4.** Validation of the (a) GIMMS LAI4g, (b) GIMMS LAI3g, (c) GLASS LAI, and (d) GLOBMAP LAI products using 113 field LAI measurements from 49 sites in the projects of BELMANIP 2.1, DIRECT 2.1, and ORNL. Sites of different vegetation biome types are marked by colors. The error metrics are $R^2$, RMSE ($m^2/m^2$), MAE ($m^2/m^2$), and MAPE (%). The blue fitting lines and dashed 1:1 lines are

drawn.

       Figure 5 shows the validation results for the four global LAI products using the remaining 20% Landsat sample points. In general, GIMMS LAI4g ($R^2 = 0.96$, RMSE = 0.32 $m^2/m^2$, MAE = 0.16 $m^2/m^2$, MAPE = 13.6%) had the highest accuracy, followed by GIMMS LAI3g ($R^2 = 0.92$, RMSE = 0.47 $m^2/m^2$, MAE = 0.26 $m^2/m^2$, MAPE = 22.2%) and GLASS LAI ($R^2 =$

0.91, RMSE = 0.50 $m^2/m^2$, MAE = 0.29 $m^2/m^2$, MAPE = 24.2%). GLOBMAP LAI ($R^2$ = 0.77, RMSE = 0.84 $m^2/m^2$, MAE = 0.46 $m^2/m^2$, MAPE = 39.1%) had the lowest accuracy. The MAPE value of 13.6% in GIMMS LAI4g achieves the LAI accuracy target proposed by GCOS.

The GIMMS LAI4g product also outperformed the other three regarding individual vegetation biome types (Figure 5). The most accurate vegetation biome varied with error metrics for all LAI products. In the GIMMS LAI4g, GIMMS LAI3g,
and GLOBMAP LAI products, SHR had the highest accuracies in $R^2$, RMSE, and MAE ($R^2$ = 0.91, 0.74, and 0.55, respectively; RMSE = 0.08, 0.14, and 0.25 $m^2/m^2$, respectively; MAE = 0.05, 0.09, and 0.20 $m^2/m^2$, respectively); meanwhile, EBF had the highest accuracies in MAPE (MAPE = 4.0%, 10.4%, and 24.2%, respectively). The most accurate vegetation biome in GLASS LAI could be DBF, SHR, or EBF, determined by $R^2$, RMSE/MAE, or MAPE, respectively. This discrepancy was attributed to the nature of the error metrics. For instance, EBF with higher absolute LAI values generally produced the lowest MAPE.
However, the $R^2$, RMSE, and MAE proposed that EBF could be the most inaccurate vegetation biome ($R^2$ = 0.55, 0.06, 0.20, and 0.05, respectively; RMSE = 0.40, 0.81, 0.89, and 1.97 $m^2/m^2$, respectively; MAE = 0.23, 0.61, 0.78, and 1.42 $m^2/m^2$, respectively). The LAI accuracy in EBF was low because it is primarily distributed in tropical areas where the quality of remote sensing data is poor owing to frequent clouds and rains.

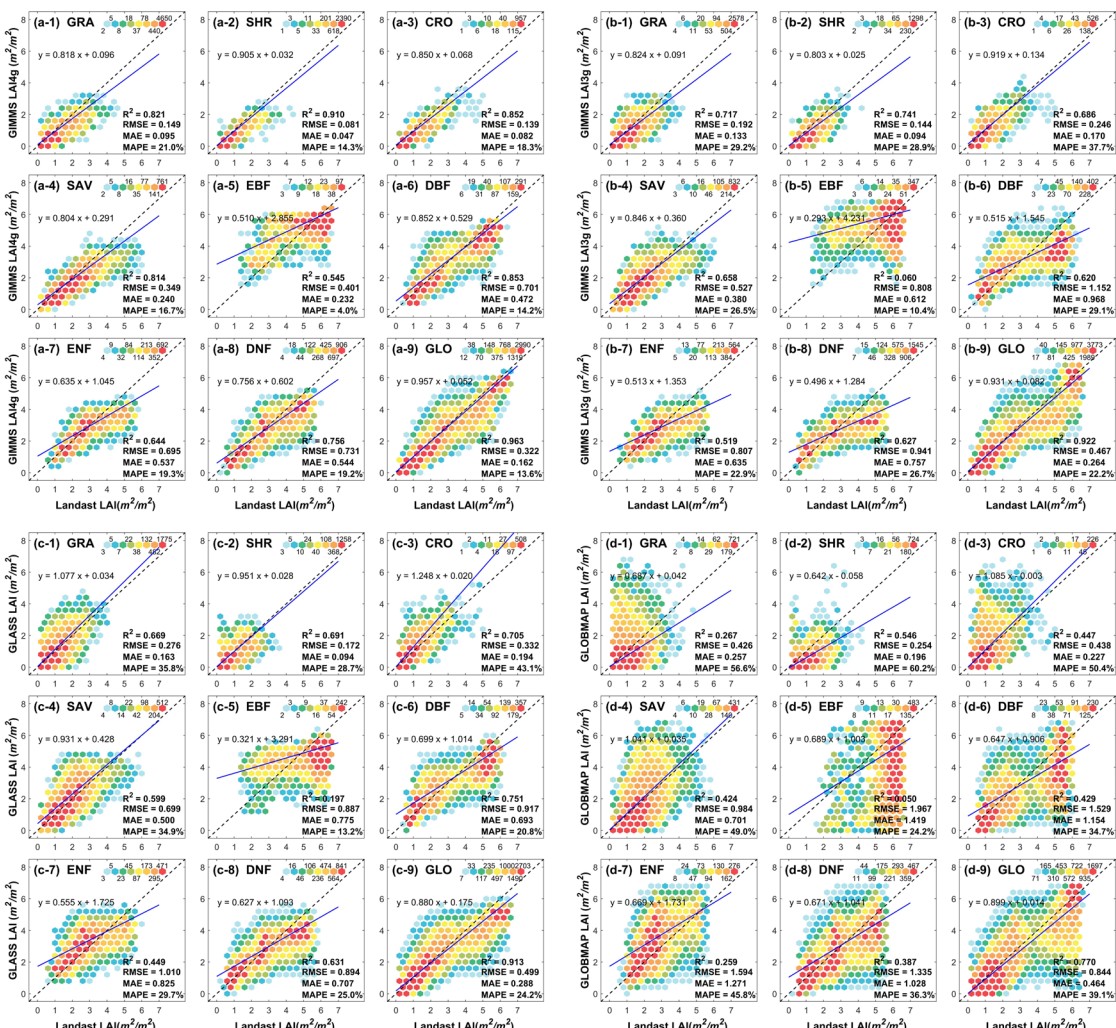

**Figure 5.** Validation of the (a) GIMMS LAI4g, (b) GIMMS LAI3g, (c) GLASS LAI, and (d) GLOBMAP LAI products in different vegetation biomes using Landsat LAI samples from 1984 to 2015. The error metrics are R$^2$, RMSE ($m^2/m^2$), MAE ($m^2/m^2$), and MAPE (%). GLO represents the global vegetation biome. The color of the dots represents LAI value frequencies in a 0.5 ($m^2/m^2$) interval.

Compared to Landsat LAI samples, the four global LAI products were underestimated in almost all the vegetation types except for CRO of GLASS LAI (Figure 5). We found certain levels of saturation in GIMMS LAI4g and GIMMS LAI3g, such as for high values of GRA, SHR, CRO, and SAV and medium values of EBF, DBF, ENF, and DNF (Figure 5). This could be attributed to the use of NDVI data in LAI models. However, we also observed the saturation effect in EBF, DBF, ENF, and DNF of GLASS LAI, which was not derived from NDVI data. For GIMMS LAI4g, the saturation was relatively subtle at a

majority of sample locations (red dots) and was obvious for locations whose LAI values deviated from the average (yellow

and blue dots). The LAI fitting line of the global vegetation biome in GIMMS LAI4g (Figure 5a-9) was close to the 1:1 line.

Figure 6 shows the dominance map of global LAI products composited by reciprocal averages of MAE from GIMMS LAI4g (green), GIMMS LAI3g (red), and GLASS LAI (blue). Grids with a small Landsat LAI sample size (< 100) were excluded as they may not provide a reliable evaluation. Within each valid 2° × 2° grid, the color was determined by the LAI products with a lower MAE, i.e., a higher absolute LAI accuracy. An immediate observation from

Figure 6 is that the absolute LAI accuracy of GIMMS LAI4g was significantly higher than others in most parts of the world. However, this advantage was relatively weak in the northern latitudes of the Eurasian continent (40°−60°). The GIMMS LAI3g and GLASS LAI could show higher accuracy at rather random locations. We acknowledged that the number of Landsat LAI samples in certain 2° × 2° grids might not be sufficient for robust accuracy assessment, but that would not alter the overall outperformance of the GIMMS LAI4g.

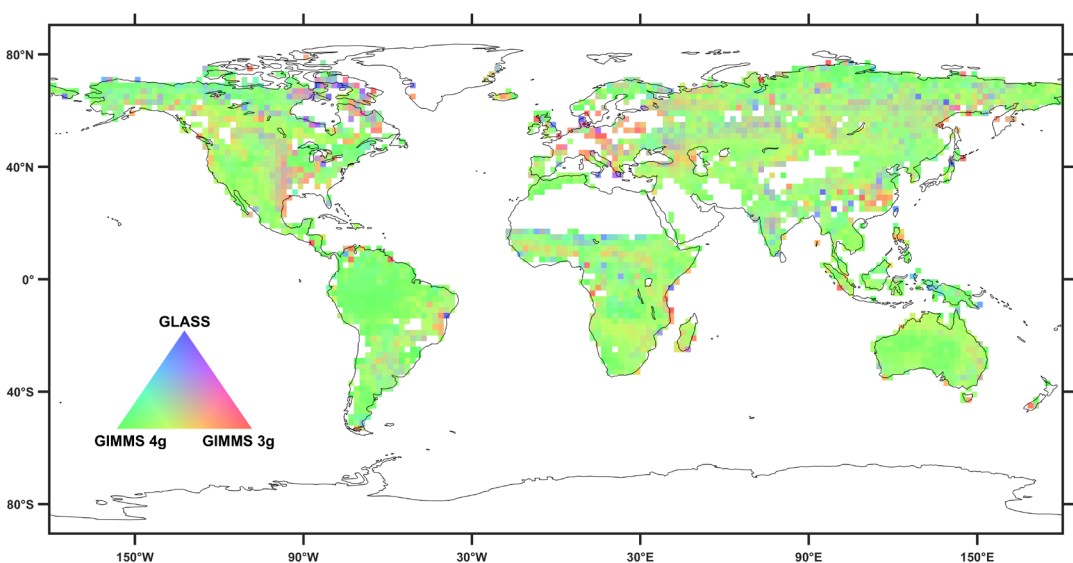

**Figure 6.** Dominance map of the GIMMS LAI3g, GIMMS LAI4g, and GLASS LAI based on their MAE. The map was drawn in 2° × 2° grids whose colors were composed of reciprocal averages of MAE from the GIMMS LAI4g (green), GIMMS LAI3g (red), and GLASS LAI (blue). Non-vegetated grids and grids with small Landsat LAI sample size (< 100) were filled white. A greener grid, for example, indicates that the GIMMS LAI4g has a lower MAE (or a higher absolute LAI accuracy).


Figure 7 showcases the spatially averaged LAI along latitude in January and July of the years 1990, 2000, and 2010 for GIMMS LAI4g, GIMMS LAI3g, GLASS LAI, and GLOBMAP LAI, respectively. The four LAI products were overall consistent. The GIMMS LAI4g and GIMMS LAI3g had lower values at northern middle latitudes (35°N – 65°N) in July

(Figure 7b; Figure 7d; Figure 7f). Also in July, the GLOBMAP LAI and GLASS LAI in the Northern Hemisphere maintained good consistency for the years 1990 and 2010 (Figure 7b; Figure 7f), but the GLOBMAP LAI was systematically lower than GLASS LAI for the year 2000 (Figure 7d). The global distribution maps of LAI in January and July can be found in Figure S4−S6.

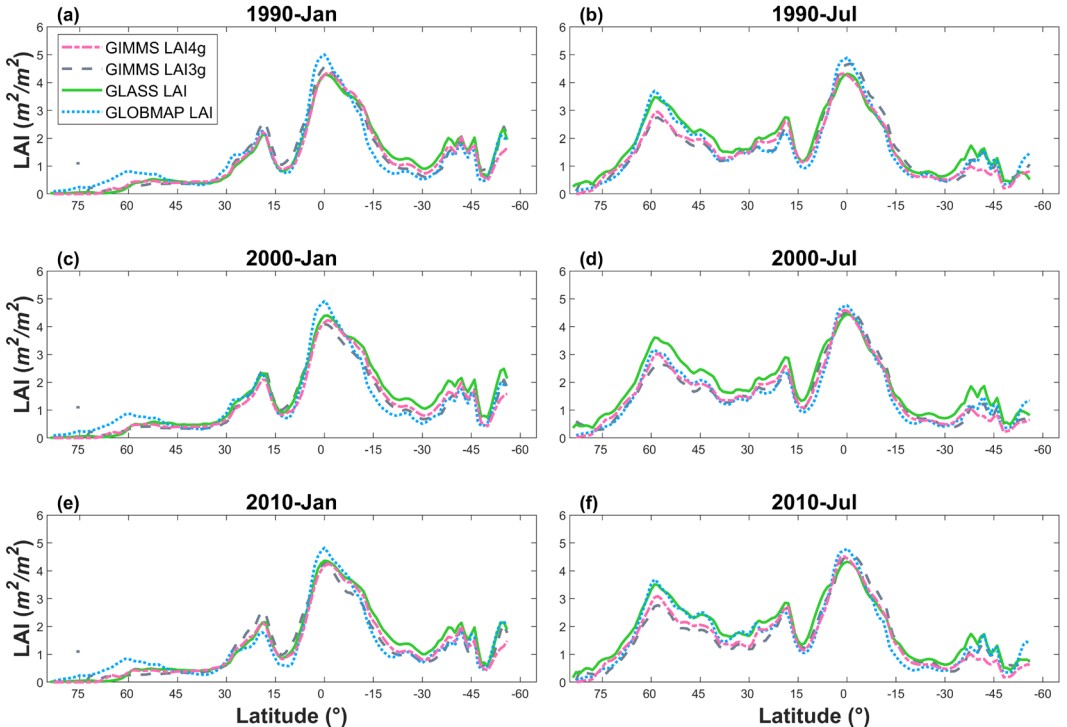

**Figure 7.** Inter-comparison of spatially averaged LAI along latitude between the GIMMS LAI4g, GIMMS LAI3g, GLASS LAI, and GLOBMAP LAI in January and July of the years 1990, 2000, and 2010. The spatial average was calculated at an interval of 1°.

## 4.4 Temporal consistency analysis

Figure 8 shows the variations of LAI bias in EBF for the GIMMS LAI4g and the other three LAI products. The LAI bias during different NOAA satellite missions was distinguished. The GIMMS LAI4g demonstrated an outstanding temporal consistency with minimum bias variations (Figure 8a), indicating an efficient removal of satellite orbital drift and sensor degradation effects. LAI bias significantly fluctuated in the GIMMS LAI3g, GLASS LAI, and GLOBMAP LAI with different patterns. The GIMMS LAI3g relied on AVHRR data only and its bias varied with NOAA missions. The evident AVHRR degradation after the year 2012, as argued by Wang et al. (2022) can be also observed in our results (Figure 8b). The GLASS LAI and GLOBMAP LAI used different data sources before (AVHRR) and after (MODIS) in the year 2000. For the GLASS LAI, bias variations before 2000 were much larger than those after 2000 (Figure 8c). The reason is likely that the data quality

from MODIS is better than AVHRR. For the GLOBMAP LAI, however, bias variations remained large for all periods of NOAA missions (Figure 8d). Similar results were found in other vegetation biome types (Figure S7-S13).

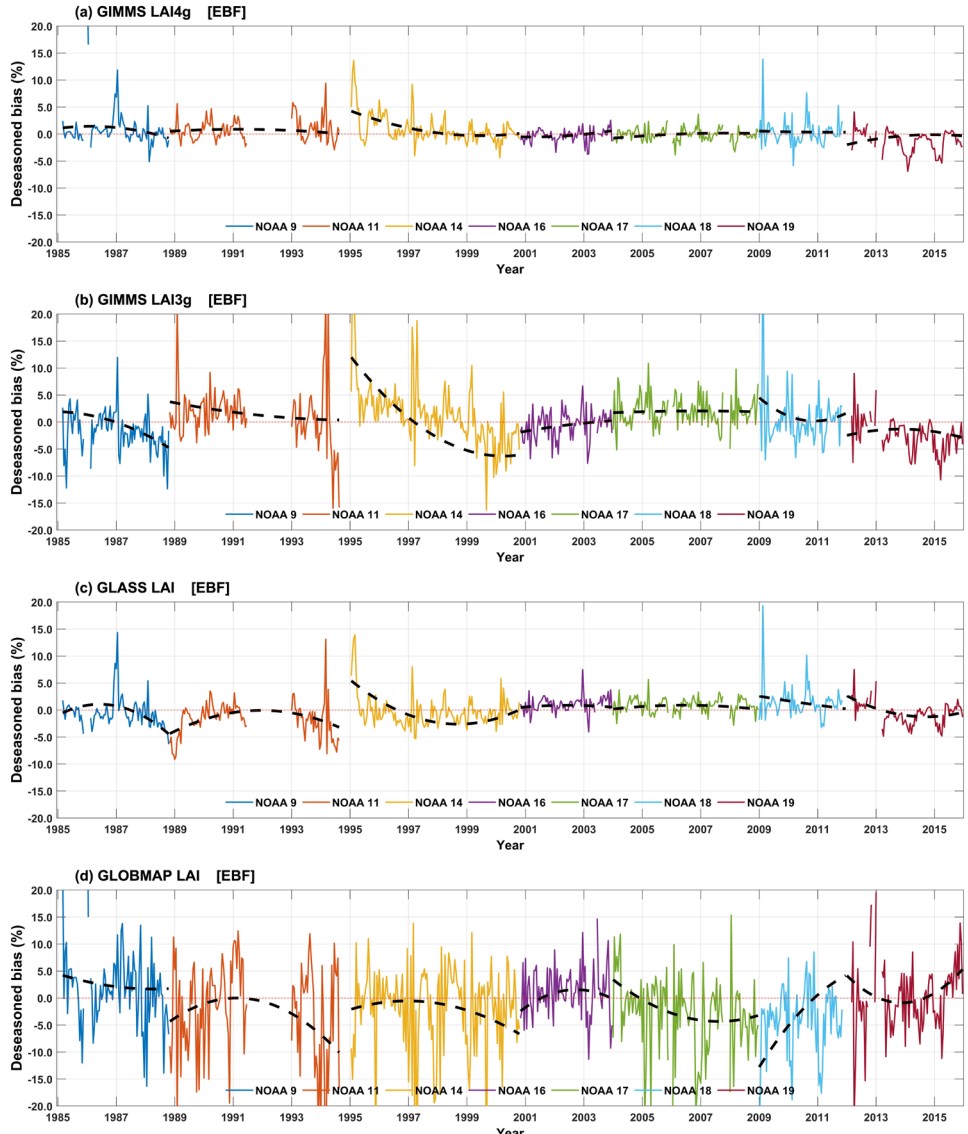

**Figure 8.** Temporal variations of LAI bias% in EBF for (a) the GIMMS LAI4g, (b) GIMMS LAI3g, (c) GLASS LAI, and (d) GLOBMAP LAI. The black dash line represents the interannual trend extracted by the EEMD method. Values from different NOAA satellite missions

were distinguished with colors.

        Figure 9 shows the GIMMS LAI4g time series before (thin black line) and after (bold colored line) data consolidation at the global scale and in selected hotspot regions of Europe, Amazon, Congo, India, and China from 1982 to 2020. The

GIMMS LAI4g shares the same footprint with the Reprocessed MODIS LAI after the year 2004. Before consolidation, there
was a systematic deviation between the GIMMS LAI4g and MODIS LAI products during 2004−2015 in all regions. The pixel-
wise fusion method has successfully matched the GIMMS LAI4g time series with MODIS LAI, eliminating abnormal shifts
in vegetation phenology. This is especially true for the Amazon rainforests, where the phenological curve has been substantially
corrected and enhanced by the MODIS LAI (Figure 9d). As a result, the temporal variations of the GIMMS LAI4g after
consolidation were self-consistent in all periods. This temporal consistency has also been evaluated regarding the vegetation
biome type and similar results were found (Figure S14).

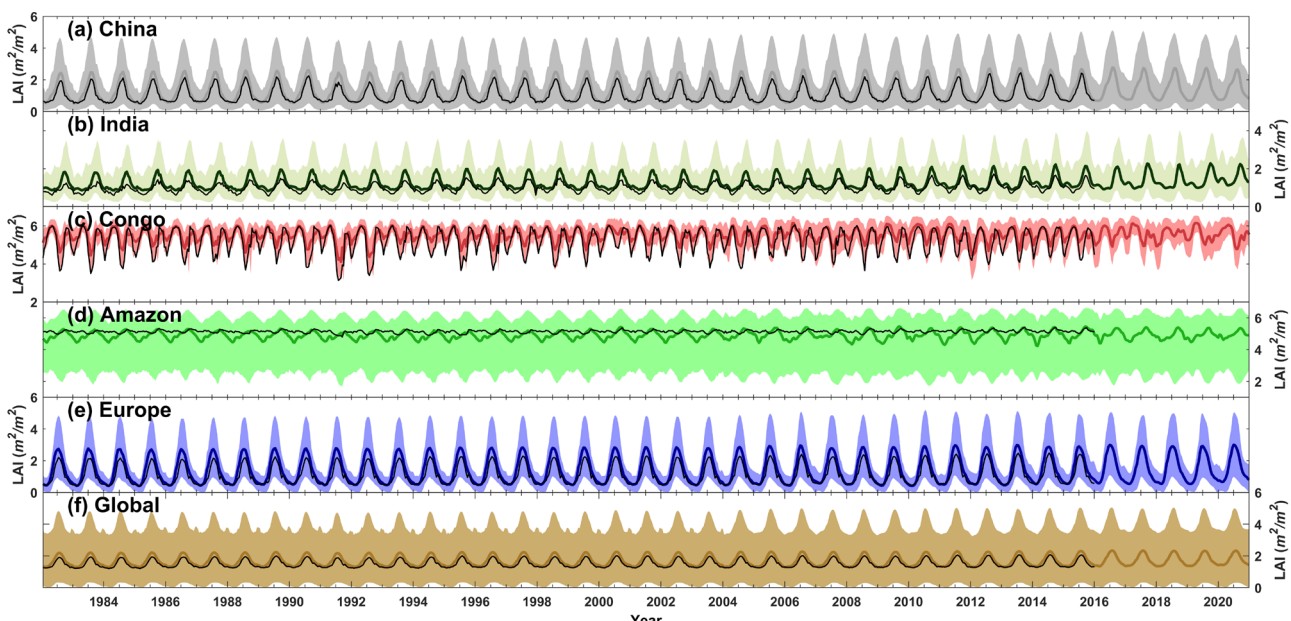

**Figure 9.** Temporal variations of the GIMMS LAI4g during 1982−2020 in selected hotspot regions of China (a), India (b), Congo (c),
Amazon (d), and Europe (e) and at the global scale (f). GLO represents the global vegetation biome. The bold colored line represents the
LAI average of GIMMSLAI4g after data consolidation, with shadow covering the value range between 10% and 90% quantiles. The thin
black line represents the LAI average of GIMMSLAI4g before consolidation. It should be noted that the GIMMS LAI4g after consolidation
shared the same footprint with the Reprocessed MODIS LAI after the year 2004.

Figure 10 shows the LAI accuracies in three periods, i.e., 1984−2015 (p1), 1984−2000 (p2), and 2001−2015 (p3), for
the four global LAI products. The results show good temporal consistency for the GIMMS LAI4g and GIMMS LAI3g. Their
accuracy differences between p2 and p3 (i.e., 1984−2000 and 2001−2015) were minimal for most vegetation biomes. In
particular, the global vegetation biome shows constant $R^2$ values (GIMMS LAI4g: 0.96 (p2) vs. 0.96 (p3); GIMMS LAI3g:
0.92 vs. 0.92) (Figure 10a) and a small difference in RMSE (GIMMS LAI4g: 0.31 $m^2/m^2$ vs. 0.34 $m^2/m^2$; GIMMS LAI3g: 0.45
$m^2/m^2$ vs. 0.48 $m^2/m^2$) (Figure 10b), MAE (GIMMS LAI4g: 0.15 $m^2/m^2$ vs. 0.17 $m^2/m^2$; GIMMS LAI3g: 0.25 $m^2/m^2$ vs. 0.27

$m^2/m^2$) (Figure 10c), and MAPE (GIMMS LAI4g: 13.72% vs. 13.53%; GIMMS LAI3g: 22.67% vs. 21.85%) (Figure 10d).

The temporal consistency of GLOBMAP LAI in different periods was relatively low. For the GLASS LAI and GLOBMAP LAI that used different data sources before (AVHRR) and after (MODIS) the year 2000, data quality after 2000 was higher than that before 2000 because of the improvement in satellite sensors. The GIMMS LAI4g product used Landsat LAI samples that covered the whole period from 1984 to 2015. This consistency in LAI reference data resulted in a minimum difference between periods.

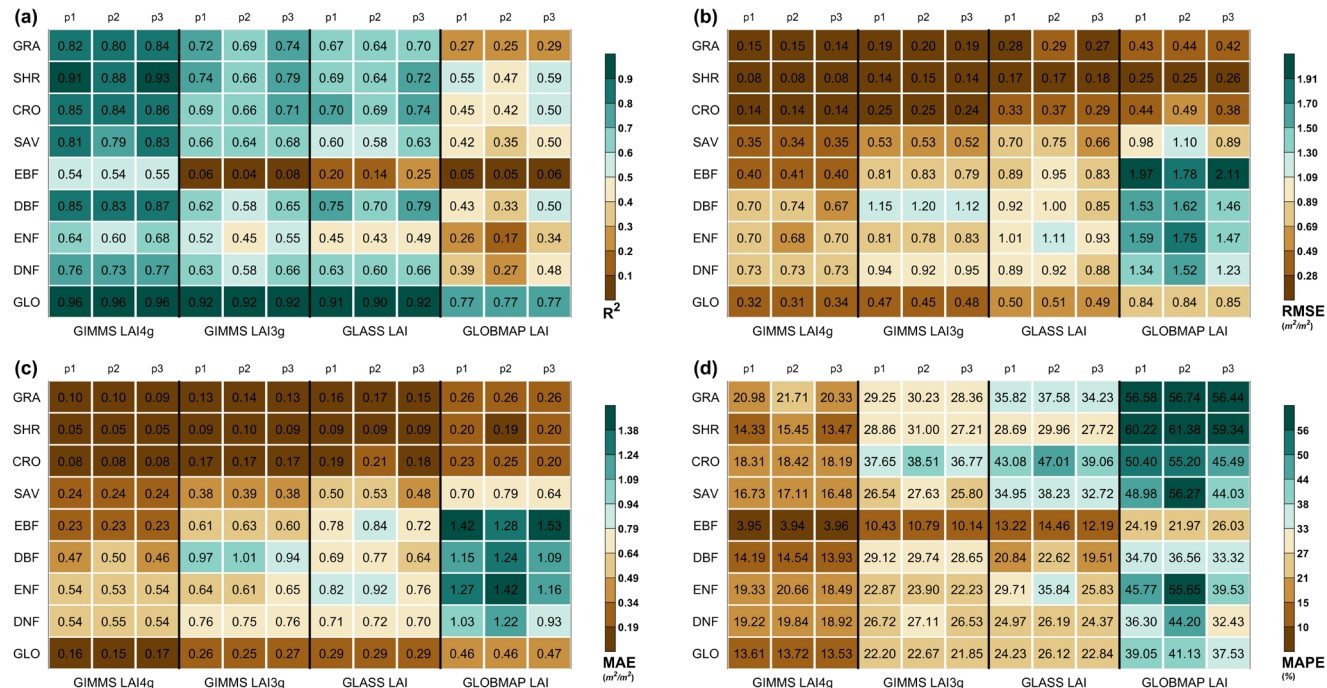

**Figure 10.** Temporal consistencies between different periods for the global LAI products. The global LAI products include GIMMS LAI4g, GIMMS LAI3g, GLASS LAI, and GLOBMAP LAI). The periods are 1984−2015 (p1), 1984−2000 (p2), and 2001−2015 (p3). The consistencies were evaluated at the biome level using $R^2$ (a), RMSE (b), MAE (c), and MAPE (d) calculated based on Landsat LAI samples. GLO represents the global vegetation biome.


Figure 11 demonstrates a good consistency in the overlapping periods between annual variations of the final GIMMS LAI4g product (GIMMS LAI4g after consolidation) and the input and intermediate products. The shapes of the anomalies were similar. The LAI trends for GIMMS LAI4g remained consistent before and after consolidation (2.2 ×10⁻³ $m^2m^{-2}yr^{-1}$ vs. 2.4 ×10⁻³ $m^2m^{-2}yr^{-1}$), despite Reprocessed MODIS LAI presenting a highly greening trend during 2004−2020 (5.6 ×10⁻³ $m^2m^{-2}yr^{-1}$).

The consistency has also been found in a biome-specific manner (Figure S15). This result indicates that both BPNN modeling (PKU GIMMS NDVI vs. GIMMS LAI4g before consolidation) and data consolidation (GIMMS LAI4g before consolidation vs. GIMMS LAI4g after consolidation) preserved the LAI anomaly and trend.

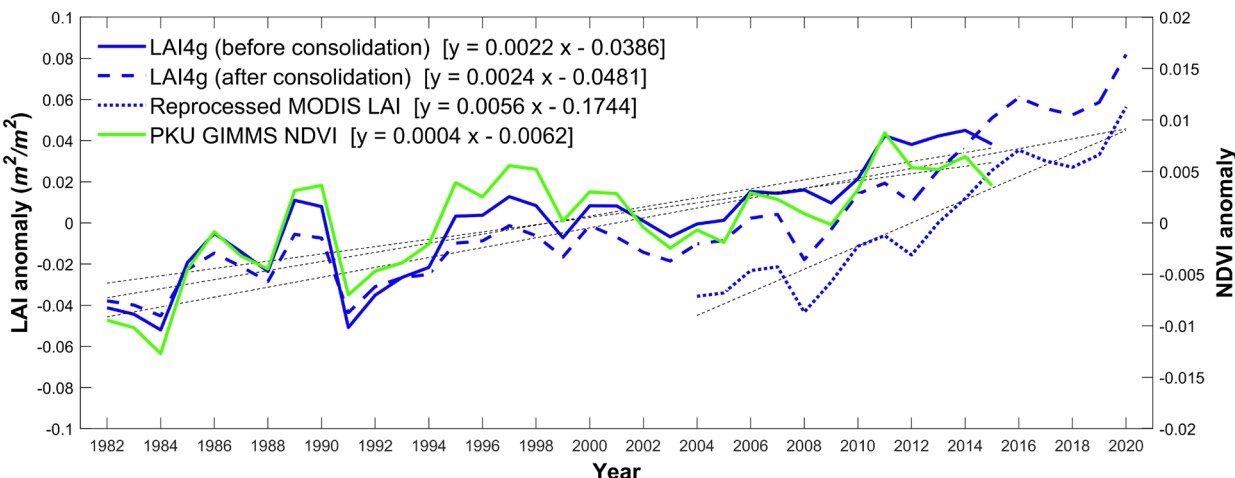

**Figure 11.** Annual anomalies and trends of GIMMS LAI4g before consolidation (1982−2015), GIMMS LAI4g after consolidation (1982−2020), Reprocessed MODIS LAI (2004−2020), and PKU GIMMS NDVI (1982−2015). Note that the regression equations within the square brackets were calculated from different periods depending on the products.

### 4.5 LAI trend analysis

Figure 12a to Figure 12d show the slope maps of the LAI time series from the GIMMS LAI4g (after consolidation), GIMMS LAI3g, GLASS LAI, and GLOBMAP in the period of 1982−2015. Figure 12e to Figure 12g shows the slope differences between the GIMMS LAI4g and the other three LAI products. In general, the GIMMS LAI4g, GIMMS LAI3g, and GLASS LAI showed a similar spatial pattern that agreed on the greening trend in global hotspot areas such as China and India. The GIMMS LAI4g demonstrated a more significant greening trend in the high-latitude regions of northern Europe and Asia.

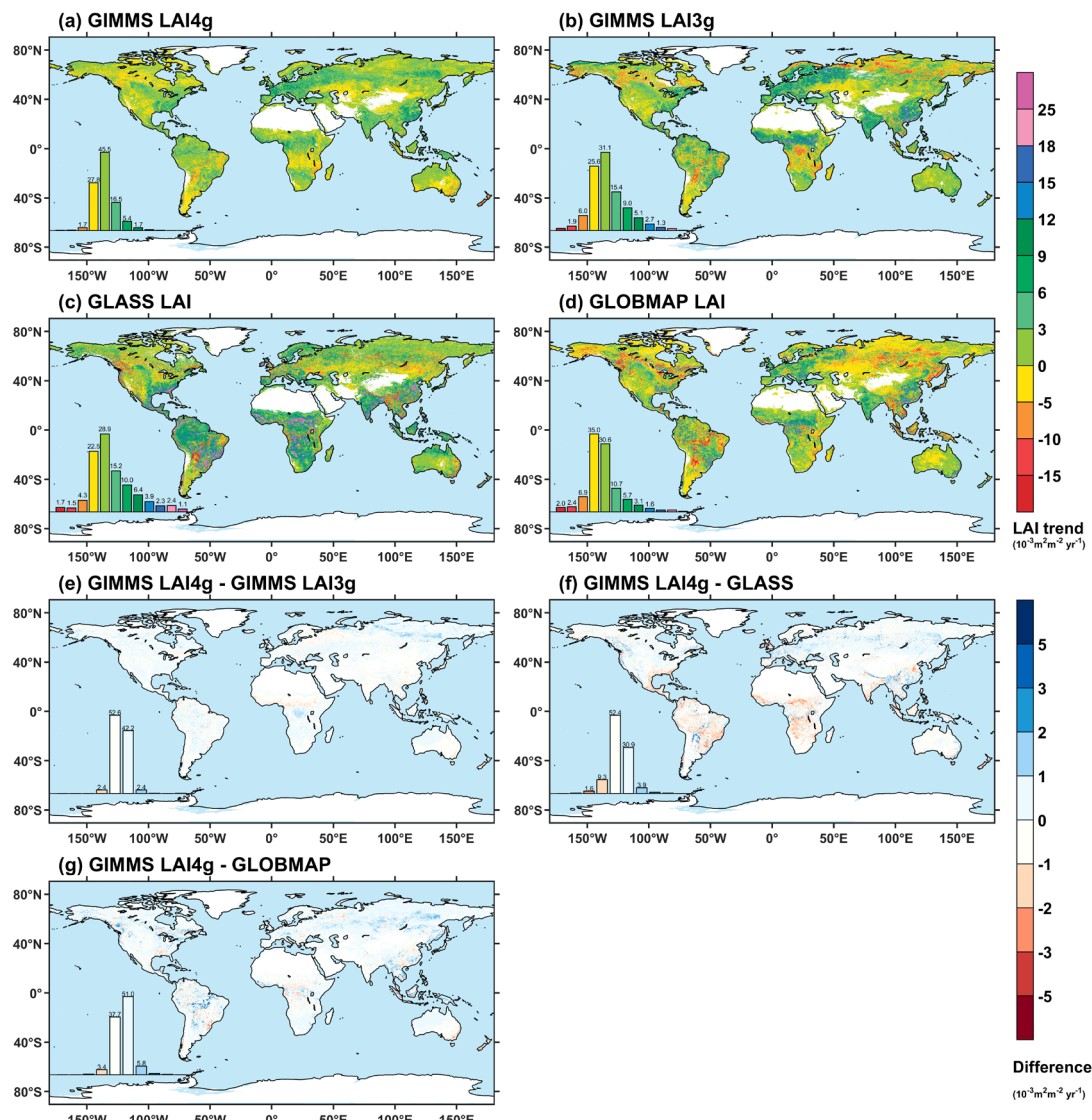

**Figure 12.** Global maps of LAI trends and their differences between the global LAI products during 1982−2015. The LAI products include GIMMS LAI4g after consolidation (a), GIMMS LAI3g (b), GLASS LAI (c), and GLOBMAP LAI (d). The trend was calculated as the slope of a linearly fitted LAI time series. (e) to (g) show the slope differences between the GIMMS LAI4g and the other three LAI products.

Figure 13a shows the annual average LAI trends during 1982−2015 (p1), 1982−2000 (p2), and 2001−2015 (p3) for different vegetation biomes of the four LAI products. For the whole period domain (1982−2015), the GIMMS LAI4g and GIMMS LAI3g products presented a similar greening trend for the global vegetation biome, with a slope value of $1.77 \times 10^{-3}$ $m^2 m^{-2} yr^{-1}$ and $2.06 \times 10^{-3}$ $m^2 m^{-2} yr^{-1}$, respectively. The greening trend was much higher in the GLASS LAI product ($3.81 \times 10^{-3}$

$m^2m^{-2}yr^{-1}$) and much lower in the GLOBMAP LAI product ($0.05\times10^{-3}$ $m^2m^{-2}yr^{-1}$). GIMMS LAI4g had the maximum global LAI trends in forest type of DNF; and its trends in other biomes were in-between the maximum and minimum trends of GIMMS LAI3g, GLASS LAI, and GLOBMAP LAI. Before 2000, the four LAI products generally demonstrated greening trends except for EBF in GIMMS LAI4g ($-0.02\times10^{-3}$ $m^2m^{-2}yr^{-1}$) and GLOBMAP LAI ($-0.51\times10^{-3}$ $m^2m^{-2}yr^{-1}$). The LAI products showed few agreements in vegetation trends after 2000. GIMMS LAI4g and GLOBMAP LAI exhibited continuous greening in all biomes, GIMMS LAI3g exhibited browning in SHR and EBF, and GLASS LAI was dominated by a browning trend in GRA, SAV, EBF, DBF, and ENF. We also paid attention to the vegetation trends in the EBF of Amazon and Congo (Figure S16). Large inconsistencies were found between the LAI products. Almost all the LAI products presented a greening trend except the GIMMS LAI3g in the Congo forests ($-4.7\times10^{-3}$ $m^2m^{-2}yr^{-1}$) and the GLOMAP LAI in the Amazon forests ($-1.8\times10^{-3}$ $m^2m^{-2}yr^{-1}$). The GIMMS LAI4g had moderate greening trends compared to other products.

Figure 13b shows the annual LAI variations of the four LAI products for different vegetation biomes. The GIMMS LAI4g demonstrated continuous global greening trends across 1982−2015; meanwhile, the GLOBMAP suffered from a noticeable decrease in trend around the year 2000. The GIMMS LAI3g and GLASS LAI showed remarkable trend differences before and after the year 2000. Their LAI values significantly increased before 2000 but turned stagnant after 2000. As both GIMMS LAI3g and GIMMS LAI4g were based on AVHRR after the year 2000, we attributed their opposite trends to the effect of AVHRR sensor degradation presented in GIMMS LAI3g (Wang et al., 2022) because MODIS LAI also showed a greening trend during this period (Wang et al., 2022; Jiang et al., 2017).

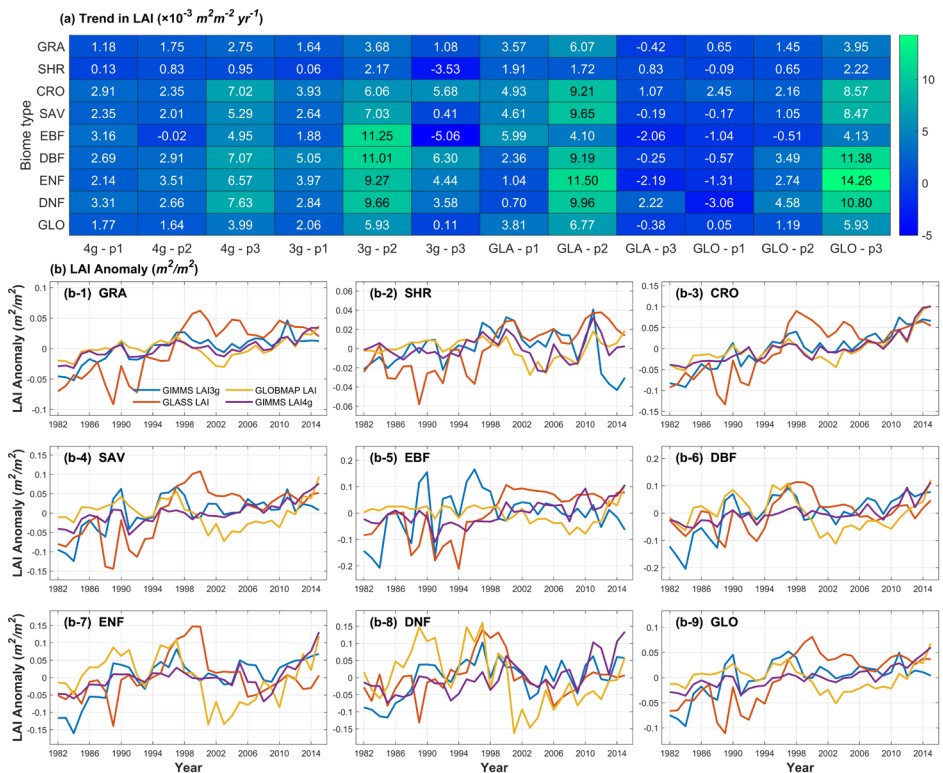

**Figure 13.** Variations of annual LAI anomaly of different vegetation biomes in the global LAI products during 1982−2015. The LAI products include GIMMS LAI4g, GIMMS LAI3g, GLASS LAI, and GLOBMAP LAI. (a) shows the slope values of the annual LAI during 1982−2015 (p1), 1982−2000 (p2), and 2001−2015 (p3). In the x-axis, 4g, 3g, GLA, and GLO stands for GIMMS LAI4g, GIMMS LAI3g, GLASS LAI, and GLOBMAP LAI, respectively. (b) shows the annual LAI time series.

## 5. Discussion

### 5.1 Improvements over other long-term global LAI products

The remote sensing data source and LAI reference data are critical inputs for the accurate derivation of long-term and large-scale LAI products. The first improvement in this study was the use of a more reliable remote sensing product, i.e., the PKU GIMMS NDVI product. Some global LAI products (e.g., GLASS LAI and GLOBMAP LAI) directly used remote sensing surface reflectance as the data source (Ma and Liang, 2022; Kang et al., 2021; Xiao et al., 2016), but some others (e.g., GIMMS LAI3g) argued that NDVI could be more robust against the terrain, atmospheric conditions, and BRDF effects (Zhu et al., 2013; Pinzon and Tucker, 2014; Zeng et al., 2022). More importantly, the PKU GIMMS NDVI product essentially addresses the issue related to the NOAA satellite orbital drift and AVHRR sensor degradation (Li et al., 2023a), which had widely existed

in current long-term global LAI products (Zhu et al., 2013) (Figure 8) since the AHVRR data were the only data source before the late 1990s that provided spatiotemporal observations over the globe. Our GIMMS LAI4g thereby was also free from this issue (Figure 8). The second improvement in our LAI product was the use of massive and high-quality Landsat LAI samples. In current long-term global LAI products (e.g., GLASS LAI, GLOBMAP LAI, and GIMMS LAI3g), the LAI reference data were either ground measurements that were spatially and temporally insufficient or LAI values derived from advanced sensors unavailable before the year 2000 (Zhu et al., 2013; Chen et al., 2019a). In the LAI model generalization, uncertainties could be hard to determine over locations and dates when the LAI reference data were basically absent. The Landsat LAI samples used in this study had a large number (3.6 million), a long time series (1984−2015), and global coverage (Figure 2). These two improvements in this study guaranteed that our GIMMS LAI4g product is more spatiotemporally consistent, as demonstrated in our results (sections 4.3 and 4.4).

Besides, incorporating other explanatory variables, including spatial, temporal, and satellite-based information, further improved the robustness of LAI models. Specifically, the role of spatial information (longitude and latitude) and temporal information (month) has been underscored in explaining global LAI variations (Figure 3 and Table 1). In this study, individual BNPP models were developed for vegetation biomes. We chose not to include the vegetation biome type as an explanatory variable (GIMMS LAI3g did) because the values of the vegetation biome type are deterministic rather than continuous (like month). The deterministic NOAA satellite number was used as an explanatory variable so that the temporal consistency of the BPNN models can be ensured.

Compared to its predecessor (GIMMS LAI3g; 1982−2016) which relied on AVHRR data only, our GIMMS LAI4g (1982−2020) provides up-to-date LAI data by consolidating with the Reprocessed MODIS LAI product. This extension of temporal coverage could help interpret recent global vegetation dynamics. Two other LAI products, namely the GLASS LAI (1982−2018) and GLOBMAP LAI (1982−2020), also incorporated MODIS data (reflectance). However, they did not explicitly calibrate systematic deviations between AVHRR and MODIS data (Liu et al., 2012; Xiao et al., 2014). Our study employed a pixel-wise fusion method to match the GIMMS LAI4g with the MODIS LAI product. The results showed an excellent consistency between the GIMMS LAI4g (after consolidation) and MODIS LAI.

## 5.2 Potential applications of the GIMMS LAI4g product

With an explicit physical meaning, LAI was proposed to be more accurate in characterizing vegetation dynamics than spectral indices such as NDVI and Enhanced Vegetation Index (EVI) (Zhang et al., 2004; Verger et al., 2016). Our results demonstrated that the GIMMS LAI4g product could be more spatiotemporally consistent and reliable than other long-term global LAI products. One important role of the GIMMS LAI4g is to mitigate the disagreements between current global LAI products and gain robust knowledge about long-term vegetation changes. For the past 40 years, the long-term analysis based on the global LAI products has shown an overall greening trend in most vegetated areas. However, significant variations existed between different LAI products at the regional scale (Wang et al., 2022; Jiang et al., 2017). In the evergreen broadleaf

forests of Africa, for instance, the GIMMS LAI3g exhibited a decreasing trend from the year 2000 while the MODIS LAI exhibited an increasing trend (Wang et al., 2022). The GIMMS LAI4g provides an opportunity to better understand the spatial pattern of vegetation greening (or browning) and its drivers (Zhu et al., 2016; Piao et al., 2015; Chen et al., 2019a).

LAI is also a popular proxy for many important ecosystem attributes and functions, such as carbon stock and sink (Chen et al., 2019b), nutrition cycle (Pierce et al., 1994), and evapotranspiration (Wang et al., 2014). It serves as a fundamental parameter in many ecosystem models (Boussetta et al., 2013; Boussetta et al., 2015; Chen et al., 2015), earth system models (Mahowald et al., 2016), and climate models (Boussetta et al., 2013; Boussetta et al., 2015). The GIMMS LAI4g is expected to benefit the development of these models and provide a powerful data basis for a more accurate and reliable land surface characterization.

## 5.3 Uncertainty sources of GIMMS LAI4g product

The PKU GIMMS NDVI product and the Landsat LAI samples comprise this study's primary sources of uncertainty. Despite the efforts by Zha et al. (2023), the number of Landsat LAI samples was small in certain regions, e.g., the northern high latitudes and tropical areas. This was attributed to the low solar altitude angle, polar night phenomenon, and climate conditions such as frequent clouds, snow, and rains at the time of Landsat observation. Also, the Landsat LAI samples were 585 absent before 1984 and scarce in 1984 (section 4.1), which would produce larger uncertainties for the GIMMS LAI4g product during the NOAA-7 period (July 1981 to February 1985). The PKU GIMMS NDVI product suffers from the same issues as it was also derived from Landsat samples (Li et al., 2023a). Although both Li et al. (2023a) and the current study have used MODIS data as compensation, the relative lack of Landsat samples in certain regions and time may still lower the robustness of models in the generation of GIMMS LAI4g. The use of PKU GIMMS NDVI could also result in the saturation effect in 590 GIMMS LAI4g as NDVI data tend to saturate at high values (Figure 5). This study established biome-specific models that incorporate multiple explanatory variables besides PKU GIMMS NDVI to account for the LAI variations in space, time, biome, and satellite. This effort could help alleviate the saturation effect though the effect still exists.

In addition, this study used a static global land cover map determined by the most frequent biome type within each grid between 2001 and 2019. This strategy could bring potential uncertainties yet represents a balance between the sample size and 595 sample quality for GIMMS LAI4g generation and validation. When applying GIMMS LAI4g for vegetation trends analysis, a careful consideration of land cover change is suggested. Other sources of uncertainty could be from the BPNN model structure in which more explanatory variables such as temperature and precipitation could be incorporated, and the Reprocessed MODIS LAI product. It should be noted that, however, these uncertainties also existed in other LAI products and this study has tried its best to mitigate their influence on the GIMMS LAI4g.

# 6. Conclusions

This study developed a new generation of the GIMMS LAI product (GIMMS LAI4g, 1982−2020) based on Back Propagation Neural Network (BPNN) models and a pixel-wise consolidation method. The GIMMS LAI4g was featured by the use of the PKU GIMMS NDVI product and the massive high-quality Landsat LAI samples. The recently published PKU GIMMS NDVI efficiently removed the effects of NOAA orbital drift and AVHRR sensor degradation, which has been a critical issue in existing LAI products. The high-quality global Landsat LAI samples, with a total number of 3.6 million and temporal coverage of 1984−2015, facilitated the creation of spatiotemporally consistent BPNN models. The spatiotemporally consistent GIMMS LAI4g product covers a time span of 1982 to 2020, with a spatial resolution of 1/12° and a temporal resolution of half-month. It can potentially provide strong data support for long-term vegetation monitoring and model development with high accuracy and reliability, as shown below:

- Evaluated by the Landsat LAI samples, the GIMMS LAI4g product ($R^2$=0.96, RMSE = 0.32 $m^2/m^2$, MAE=0.16 $m^2/m^2$, MAPE=13.6%) was overall more accurate than the mainstream global LAI products, including the GIMMS LAI3g ($R^2$=0.92, RMSE = 0.47 $m^2/m^2$, MAE=0.26 $m^2/m^2$, MAPE=22.2%), GLASS LAI ($R^2$=0.91, RMSE = 0.50 $m^2/m^2$, MAE=0.29 $m^2/m^2$, MAPE=24.2%), and GLOBMAP LAI ($R^2$=0.77, RMSE = 0.84 $m^2/m^2$, MAE=0.46 $m^2/m^2$, MAPE=39.1%). Its accuracy meets the target proposed by the Global Climate Observation System (GCOS).

- Evaluated by field LAI measurements, GIMMS LAI4g ($R^2$ = 0.70, RMSE = 0.86 $m^2/m^2$, MAE = 0.60 $m^2/m^2$, MAPE = 32.8%) had comparable accuracies with GIMMS LAI3g ($R^2$ = 0.72, RMSE = 0.78 $m^2/m^2$, MAE = 0.56 $m^2/m^2$, MAPE = 30.4%) and GLASS LAI ($R^2$ = 0.68, RMSE = 0.83 $m^2/m^2$, MAE = 0.60 $m^2/m^2$, MAPE = 32.8%) and lowest underestimation among all global long-term LAI products.

- The GIMMS LAI4g outperformed the other LAI products in most regions of the globe and all vegetation biomes ($R^2$: 0.55 to 0.91; RMSE: 0.08 $m^2/m^2$ to 0.73 $m^2/m^2$; MAE: 0.05 $m^2/m^2$ to 0.54 $m^2/m^2$; MAPE: 4% to 21%).

- The GIMMS LAI4g product removed the effects of NOAA orbital drift and AVHRR sensor degradation, which can be observed in other LAI products.

- The GIMMS LAI4g after consolidation with the Reprocessed MODIS LAI was more temporally consistent between the three periods of 1984−2015, 1984−2000, and 2001−2015 than other LAI products. It more reasonably depicted global vegetation trends (greening or browning) and demonstrated a continuous global greening trend before and after 2000.

**Data availability**

The spatiotemporally consistent global dataset of the GIMMS Leaf Area Index (GIMMS LAI4g) generated in this study is openly available at https://doi.org/10.5281/zenodo.7649107 (Cao et al., 2023). It covers the whole global vegetation area at a half-month temporal resolution and 1/12° spatial resolution from 1982 to 2020. It is available in Geographic Lat/Lon projection and TIFF format. In the same repository, we have also provided the version of GIMMS LAI4g that is solely based on AVHRR data, which means that its generation was free from the consolidation with Reprocessed MODISL LAI and it used the version of PKU GIMMS NDVI before consolidation with MODIS NDVI. Before applying the GIMMS LAI4g product, we highly recommend users read the Readme file in the repository and properly handle the fill value and the quality control flag in the dataset.

**Author contributions.** ZZ conceptualized and supervised the project. ZZ, ML, SC, ZW, and JZ designed the workflow of methodology to product the dataset. ML, ZW, JZ, WZ, ZD, JC, YZ, and YC conducted the work in data acquisition and processing. ZZ, ML, SC, ZW, and JZ performed data analysis. SC, ML, ZZ, RM, and SP prepared the manuscript. SC, ZZ, RM, and SP reviewed and edited the draft. All authors contributed to the interpretation of the results.

**Competing interests.** The authors declare that they have no known competing financial interests or personal relationships that could have influenced the work reported in this study.

**Disclaimer.** Publisher's note: Copernicus Publications remains neutral with regard to jurisdictional claims in published maps and institutional affiliations.

**Acknowledgments.** We would like to thank NASA for providing MODIS data products, Zhiqiang Xiao for providing GLASS LAI data, Yang Liu and Ronggao Liu for providing GLOBMAP LAI data, and Hua Yuan for providing the reprocessed MODIS LAI product. We gratefully acknowledge the Landsat data support from USGS and Google Earth Engine (https://earthengine.google.com/, last access: September 2023). We are grateful to the anonymous reviewers for their constructive comments and suggestions. We also thank Dr. Jorge Luis Peña-Arancibia and Dr. Zaved Khan from CSIRO for bringing up issues in southeast Australia during the preprint posting of the manuscript. The updated algorithm with the reviewers' suggestions has been used for generating the GIMMS LAI4g product.

**Financial support.** This work was supported by the National Natural Science Foundation of China (42271104, 41901122, 42001355), the Shenzhen Fundamental Research Program (GXWD20201231165807007-20200814213435001), and the Shenzhen Science and Technology Program (JCYJ20220531093201004).

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
