# Peer review of "Spatiotemporally consistent global dataset of the GIMMS Leaf Area Index (GIMMS LAI4g) from 1982 to 2020"

_Earth System Science Data, 2023_

## Author Response (AR1)

**To Referee #1**

Dear reviewer,

We are very pleased to finish a revised version of the manuscript essd-2023-68 entitled "**Spatiotemporally consistent global dataset of the GIMMS Leaf Area Index (GIMMS LAI4g) from 1982 to 2020**". In preparing this revision we have considered all your comments and incorporated most of the suggestions. We greatly appreciate your time and effort spent reviewing this manuscript, which has improved the revised version of the manuscript.

**Substantial improvements have been made based on your comments, including:**

(1) More details have been provided for the two datasets of PKU GIMMS NDVI and Landsat LAI samples. Specifically, statistics on their performance are now available.

(2) The GIMMS LAI4g is now validated by ground LAI measurements.

(3) The saturation issue in different vegetation biomes of GIMMS LAI4g has been discussed.

(4) The vegetation trends before and after 2000 have been analyzed for GIMMS LAI4g and other global long-term LAI products.

Below we provide point-to-point responses, each following the specific comment from the reviewer. All the changes in the revised manuscript have been marked in red.

Sincerely yours,

Zaichun Zhu, Ph. D. (on behalf of the author team)

School of Urban Planning and Design

Peking University

Tel: 86 185 0042 6608

Email: zhu.zaichun@pku.edu.cn

**[Comment 1]** *Cao et al. generated a new global LAI dataset GIMMS LAI4g (hereafter, LAI4g), this dataset overcome the effect of satellite orbital drift and sensor degradation from NOAA series satellites. Its results may explain the LAI trend bias between pre-MODIS era (1982-1999) and MODIS era (2000-present) and it also explains whether the vegetation area is greening or browning over the world. This study fills the gap of how the AVHRR sensors affect the LAI trend (the significant improvement is showed at figure 6), which have not been solved in the previous studies.*

**[Response 1]** We thank the referee for highlighting the significance of this research. We are happy to provide a new version of the GIMMS LAI product (GIMMS LAI4g) that eliminates the effect of NOAA satellite orbital drift and AVHRR sensor degradation. On top of that, GIMMS LAI4g also demonstrates high absolute accuracies which meet the accuracy target proposed by the Global Climate Observation System. We hope this product could distinguish the true vegetation trend from uncertainties from NOAA/AVHRR and LAI references and help better understand the vegetation dynamic under the current global environmental changes.

**[Comment 2]** *However, three major points needed to be improved before publication.*
*#1 The authors used two datasets, i.e. PKU GIMMS NDVI (Li et al. under review), Landsat LAI samples (Zha et al. in preparation) as the principal data source for generating LAI4g product. However, these two datasets have not finished their peer-review process, so these two datasets are not fully convincing. At least, the authors should show the statistics on how robust these two datasets are. I also suggested the authors adding the ground measured LAI (could be found in Xu et al. 2018 RSE and Ma et al. 2022 RSE) as validation to verify LAI4g product.*
*Xu, B., Li, J., Park, T., Liu, Q., Zeng, Y., Yin, G., ... & Myneni, R. B. (2018). An integrated method for validating long-term leaf area index products using global networks of site-based measurements. Remote Sensing of Environment, 209, 134-151.*
*Ma, H., & Liang, S. (2022). Development of the GLASS 250-m leaf area index product (version 6) from MODIS data using the bidirectional LSTM deep learning model.*

*Remote Sensing of Environment, 273, 112985.*

**[Response 2]** We thank the reviewer for these suggestions. In the revised manuscript, we have provided more details on the generation process of two datasets (PKU GIMMS NDVI and Landsat LAI samples) and the statistics of their performance in absolute accuracy and vegetation trend. The annual variation of Landsat LAI sample size has been added to the updated Figure 2. We have also provided Figure S1 in the supplementary file that uses Evergreen Broadleaf Forests (EBF) as an example to demonstrate how the effect of NOAA satellite orbital drift and AVHRR sensor degradation was eliminated in the PKU GIMMS NDVI.

   Also, a total of 113 field LAI measurements from BELMANIP 2.1, DIRECT 2.1, and ORNL were employed to evaluate the GIMMS LAI4g product as well as other global long-term LAI products. The results showed that GIMMS LAI4g ($R^2$ = 0.69, RMSE = 0.86 $m^2/m^2$, MAE = 0.61 $m^2/m^2$, MAPE = 33%) had comparable accuracies with GIMMS LAI3g ($R^2$ = 0.71, RMSE = 0.78 $m^2/m^2$, MAE = 0.55 $m^2/m^2$, MAPE = 30%) and GLASS LAI ($R^2$ = 0.68, RMSE = 0.82 $m^2/m^2$, MAE = 0.60 $m^2/m^2$, MAPE = 33%), and it had the lowest underestimation in terms of the fitting line with a slope of 0.87 and an intercept of 0.05 (Figure 4).

[revised manuscript text omitted]

**[Comment 3]** *#2 The saturation of LAI is common in LAI4g over biomes. For example, figure 3 a1 showed that the Landsat LAI ranged from 0 to 4 m2 m-2, however, the LAI4g is saturated at 3 m2 m-2. Similar conditions can be found at CRO, SHR, SVA, ENF, DNF. If there is a 1:1 line in each subplot, the underestimation of LAI4g in high range of Landsat LAI would be significant. I guess the reason the train data for LAI4g is NDVI, the NDVI may saturated but LAI not, so the model setting may lead to the LAI4g saturation.*

**[Response 3]** We agree that the saturation of NDVI at high values (dense canopy) could be propagated to downstream products. To minimize the saturation effect in the GIMMS LAI4g, we established biome-specific models that incorporate multiple explanatory variables (longitude, latitude, month, and satellite number and years since launch) besides the PKU GIMMS NDVI, to explain the LAI variations from space,

time, biome, and satellite. These explanatory variables were well contributed to the accuracies of LAI models (Figure 3). As a result, the saturation effect for a particular biome was subtle at a majority of sample locations (red dots in Figure 5) and was observable for samples whose LAI values deviated from the average (yellow and blue dots in Figure 5). For sparse vegetation types including GRA, SHR, CRO, and SAV, the GIMMS LAI4 would be underestimated at high values; and for dense vegetation types including EBF, DBF, ENF, and DNF, the GIMMS LAI4 would be underestimated at low and medium values. The 1:1 lines and fitting lines were added in the updated Figure 5. For the global vegetation biome, the LAI fitting line is close to the 1:1 line with a slope of 0.95 and an intercept of 0.07 (Figure 5a-9). It should be noted that the saturation effect can also be observed in EBF, DBF, ENF, and DNF of GLASS LAI, which was not derived from NDVI data but from surface reflectance.

Despite the saturation effect in GIMMS LAI4g and GIMMS LAI3g (based on GIMMS NDVI3g), we believed that the use of GIMMS NDVI products is beneficial as they have properly dealt with the limitations of the AVHRR instrument or operation, e.g., vicarious calibration, atmospheric and cloud correction, bias correction, satellite orbital drift, and sensor degradation (Pinzon and Tucker, 2014; Li et al., in review). The GIMMS LAI3g is one of the core data references in the IPCC Sixth Assessment Report for the assessment of global vegetation changes (Eyring et al., 2021), and has been cited for more than 600 times (from *Web of Science*).

[Figure]

**Figure 5.** Validation of the (a) GIMMS LAI4g, (b) GIMMS LAI3g, (c) GLASS LAI, and (d) GLOBMAP LAI products in different vegetation biomes using Landsat LAI samples from 1984 to 2015. The error metrics are $R^2$, RMSE, MAE, and MAPE. GLO represents the global vegetation biome. The color of the dots represents LAI value frequencies in a 0.5 ($m^2/m^2$) interval.

The following changes are made in the revised manuscript:

(a) Results on the saturation effect in Section 4.3:

"Compared to Landsat LAI samples, the four global LAI products were underestimated in almost all the vegetation types except for CRO of GLASS LAI (Figure 5). We found certain levels of saturation in GIMMS LAI4g and GIMMS LAI3g, such as for high values of GRA, SHR, CRO, and SAV and medium values of EBF, DBF, ENF, and DNF (Figure 5). This could be attributed to the use of NDVI data in LAI models. However, we also observed the saturation effect in EBF, DBF, ENF, and DNF of GLASS LAI, which was not derived from NDVI data. For GIMMS

LAI4g, the saturation was relatively subtle at a majority of sample locations (red dots) and was obvious for locations whose LAI values deviated from the average (yellow and blue dots). The LAI fitting line of the global vegetation biome in GIMMS LAI4g (Figure 5a-9) was close to the 1:1 line." (Page 17, Line 370-376)

(b) Discussions on the saturation effect in Section 5.3:

"The use of PKU GIMMS NDVI could also result in the saturation effect in GIMMS LAI4g as NDVI data tend to saturate at high values (Figure 5). This study established biome-specific models that incorporate multiple explanatory variables besides PKU GIMMS NDVI to account for the LAI variations in space, time, biome, and satellite. This effort could help alleviate the saturation effect though the effect still exists." (Page 27, Line 555-558)

**[Comment 4]** *The analysis of LAI trend can be expanded. Zhu et al. 2021 showed that the CO2 fertilization effect (CFE) is diversity from satellite LAI products, and they showed different patterns before and after 2000. Since the LAI4g overcome the effect of satellite orbital drift and sensor degradation from NOAA series satellites, I think it could solve the problem raised in Zhu et al. 2021. So, the authors can add the analysis on how the LAI trend change before and after 2000 for LAI4g, which benefits the ecology research community on how the vegetation area is greening or browning over the world.*

*Zhu, Z., Zeng, H., Myneni, R. B., Chen, C., Zhao, Q., Zha, J., ... & MacLachlan, I. (2021). Comment on "Recent global decline of CO2 fertilization effects on vegetation photosynthesis". Science, 373(6562), eabg5673.*

**[Response 4]** As suggested by the reviewer, Figure 13 has been augmented in the revised manuscript to illustrate the vegetation trends before and after 2000. Before 2000, GIMMS LAI4g demonstrated generally greening trends except for EBF ($-1.65 \times 10^{-3}$ $m^2 m^{-2} yr^{-1}$), although EBF sustained greening since 1991. After 2000, GIMMS LAI4g exhibited continuous greening for all vegetation biome types.

[Figure]

**Figure 13.** Variations of annual LAI anomaly of different vegetation biomes in the global LAI products during 1982−2015. The LAI products include GIMMS LAI4g, GIMMS LAI3g, GLASS LAI, and GLOBMAP LAI. (a) shows the slope values of the annual LAI during 1982−2020 (p1), 1982−2000 (p2), and 2001−2020 (p3). In the x-axis, 4g, 3g, GLA, and GLO stands for GIMMS LAI4g, GIMMS LAI3g, GLASS LAI, and GLOBMAP LAI, respectively. (b) shows the annual LAI time series.

The following changes are made in the revised manuscript:

(a) Results on vegetation trends before and after 2000 in Section 4:

"Figure 13a shows the annual average LAI trends during 1982−2020 (p1), 1982−2000 (p2), and 2001−2020 (p3) for different vegetation biomes of the four LAI products. For the whole period domain (1982−2020), the GIMMS LAI3g, GIMMSLAI4g, and GLASS LAI products presented a similar greening trend for the global vegetation biome, with a slope value of $2.06 \times 10^{-3}$ $m^2 m^{-2} yr^{-1}$, $2.86 \times 10^{-3}$ $m^2 m^{-2} yr^{-1}$, and $3.81 \times 10^{-3}$ $m^2 m^{-2} yr^{-1}$, respectively." (Page 23-24, Line 474-477)

"Before 2000, the four LAI products generally demonstrated greening trends except for EBF in GIMMS LAI4g ($-1.65 \times 10^{-3}$ $m^2 m^{-2} yr^{-1}$) and GLOBMAP LAI ($-0.51 \times 10^{-3}$ $m^2 m^{-2} yr^{-1}$). The LAI products showed few agreements in vegetation trends after 2000.

GIMMS LAI4g and GLOBMAP LAI exhibited continuous greening in all biomes, GIMMS LAI3g exhibited browning in SHR and EBF, and GLASS LAI was dominated by a browning trend in GRA, SAV, EBF, DBF, and ENF." (Page 24, Line 480-484)

"For EBF in GIMMS LAI4g, although its trend during 1982−2000 was overall negative (browning) (Figure 13a), it has sustained a greening trend since 1991 (Figure 13b-5)." (Page 24, Line 488-489)

**[Comment 5]** *Since the line number is not covered every line, so I just give a range of line number here.*

**[Response 5]** We feel sorry about this inconvenience. However, we used the template provided by ESSD so there is not too much we can do right now.

**[Comment 6]** *~L45 'However, the accuracies of the current LAI products have been limited by uncertainties primarily in the remote sensing data and the LAI reference data (Fang et al., 2019).' Jiang et al. 2017 GCB is also an important reference here.*

**[Response 6]** The work from Jiang et al. (2017) has been cited in the revised manuscript. The revised sentence is now "However, the accuracies of the current LAI products have been limited by uncertainties primarily in the remote sensing data and the LAI reference data (Fang et al., 2019; Jiang et al., 2017)." (Page 2, Line 44-45)

**[Comment 7]** *~ L70 to 75 It is not common to cite two papers in under review process in the introduction section. So, the authors may consider citing other published papers.*

**[Response 7]** We totally understand that the datasets in review (PKU GIMMS NDVI and Landsat LAI samples) would raise questions about how the datasets were generated and their accuracies. However, no other papers on the two datasets are currently published. We would like to claim that it is our duty to immediately cite the final version of the datasets once they finish the peer review. As compensation and as the reviewers suggested, we have added more details in the revised manuscript to

better describe PKU GIMMS NDVI and Landsat LAI samples (Section 2.1 and Section 2.2). We would also like to let the reviewer know that, the manuscript related to PKU GIMMS NDVI is currently in revision as well. All the revisions have been incorporated into the updated version of GIMMS LAI4g, despite the revisions being subtle and no major conclusions altered.

**[Comment 8]** *Section 2.1 and 2.2, these two datasets are not finished peer reviewed especially the Landsat LAI sample dataset. To ensure the GIMMS LAI4g is reliable, the authors should give enough evidence to show these two data sources are robust.*

**[Response 8]** Thanks for this comment. More details on the two datasets (PKU GIMMMS NDVI and Landsat LAI samples), including their performance in absolute accuracy and vegetation trend have been provided in Sections 2.1 and 2.2 of the revised manuscript. Please refer to **Response 2**.

**[Comment 9]** *Section 2.3 I found the authors have not used the ground measurement LAI as directly validate for LAI4g product. I recommended that the authors may use the ground measured LAI dataset in Xu et al. 2018 RSE and Ma et al. 2021 RSE to validate the LAI4g product.*

**[Response 9]** This comment is also related to **Comment 2**. We have used field LAI measurements from BELMANIP 2.1, DIRECT 2.1, and ORNL to evaluate the GIMMS LAI4g product as well as other global long-term LAI products.

**[Comment 10]** *Section 2.5 There are two version of GLASS-LAI v50, one is totally based on AVHRR data (http://www.glass.umd.edu/LAI/AVHRR/), this dataset didn't use the MODIS data as data source. The other one is fully based on MODIS with 500m spatial resolution. From my experience, there are no such dataset used both AVHRR and MODIS data as input to generate GLASS LAI product, so the authors should ensure this point.*

**[Response 10]** After a careful examination, we need to clarify that the GLASS LAI product used in this study belongs to version 4 (Xiao et al., 2016) rather than version

5. We apologize for this mistake, which has been corrected in the revised manuscript. At the time that our project started, GLASS LAI version 5 or newer was not available (Liang et a., 2021; Ma and Liang, 2022). The availability information of the latest GLASS LAI product has also been added in the revised manuscript. Please understand that with new versions of the global long-term LAI products being constantly available, it would be difficult to always use the latest versions in the study. More importantly, we believe that the update of other LAI products would not lower the significance of our GIMMS LAI4g.

The following changes are made in the revised manuscript:

(a) Introduction to GLASS LAI in Section 2.5:

"The GLASS LAI (version 4) with a temporal resolution of …" (Page 6, Line 151)
"The GLASS LAI (V4) product was acquired from ftp://ftp.glcf.umd.edu/. It should be noted that version 5 and version 6 of the GLASS LAI product have been available when our study has been prepared (Liang et a., 2021; Ma and Liang, 2022)." (Page 6, Line 156-158)

(b) Clarification in Section 3.1:

"For example, a typical ANN, …GLASS LAI (version 4) (Xiao et al., 2014), and GIMMS LAI3g (Zhu et al., 2013), respectively." (Page 8, Line 201-202)

**[Comment 11]** *Section 2.6 There are two version of GLOBMAP product. The previous version showed that the LAI is decreasing after 2000 (see Jiang et al. 2017 GCB figure1). The current version is GLOBMAP V3, which showed an increasing trend after 2000. So which one is the reliable one, it would be an open question. The authors may consider not using this dataset to do the inter-comparison.*

**[Response 11]** Thanks for this comment. The GLOBMAP LAI product used MODIS surface reflectance as input from 2001−now. For GLOBMAP LAI version 2, the MODIS product version was C5; and for GLOBMAP LAI version 3, the MODIS product version was C6 (10.1029/2012JG002084). The update from MODIS C5 to C6 addressed a series of known issues including the sensor degradation, which was believed to correct the analytical vegetation trend from negative to positive (Lyapustin

et al., 2014). With this being clarified, however, we agree with the reviewer that current global long-term LAI products of different versions need comprehensive evaluation. For our study, with the ultimate goal to signify the improvements of GIMMS LAI4g, we may still choose to use the latest versions of popular LAI products including GLOBMAP LAI. The latest versions were determined at the time our study has been prepared.

[**Comment 12**] *Section 3. A flow chart of how to generate LAI4g is useful.*

[**Response 12**] A workflow has been added accordingly in the revised manuscript, as below.

[Figure]

**Figure 1.** Schematic diagram of the generation and evaluation of the GIMMS LAI4g product.

[**Comment 13**] *Figure 1a I guess the SVA should be SAVanna? It would be more common to use the abbreviation SAV to represent savanna.*

[**Response 13**] Thanks for pointing this out. We have replaced all SVA with SAV through all texts and figures.

[**Comment 14**] *Figure 1c will the LAI distribution of different data source affect LAI4g? The Landsat LAI showed a peak around 0.5 m2 m-2, but other product didn't*

*show a similar pattern.*

**[Response 14]** We used Figure 1 (updated Figure 2; Figure 2 will be used hereafter) to demonstrate that the Landsat LAI samples have a good distribution in space (Figure 2a), time (Figure 2b), and value (Figure 2c). Specifically, Figure 2c shows that the value distribution of the Landsat LAI samples could overall match the global long-term LAI products. Regarding the concern from the reviewer, if the value distribution is significantly mismatched between LAI samples and LAI products, the samples would not be representative and suitable for training the LAI models for GIMMS LAI4g. In this sense, different LAI distributions will affect LAI4g. We recognize that Figure 2c was not properly drawn as the frequency curves overlapped each other. A new figure has been made in the revised manuscript. The new figure shows that both GIMMS LAI3g and Landsat LAI samples peak at 0.5 $m^2/m^2$ and the peaking values for GLASS LAI and GLOBMAP LAI were smaller.

To further demonstrate the temporal representation of Landsat LAI samples, we have also updated Figure 2b to show the inter-annual variation of the Landsat LAI sample size. During 1984−2015, the Landsat LAI sample size per year increased from 28,323 in 1984, peaked at 200,315 in 2001 when both Landsat 5 and Landsat 7 were available, and then leveled off until 2012 (sample size: 22,106). As Landsat 5's decommissioning, very few images were acquired from November 2011 to May 2012 (https://www.usgs.gov/centers/eros/science/usgs-eros-archive-landsat-archives-landsat-4-5-thematic-mapper-tm-level-1-data). Since the launch of Landsat 8 in 2013, the Landsat LAI sample became steadily available again.

[Figure]

**Figure 2.** Spatial and temporal distribution of the LAI reference data. (a) The global distribution of LAI samples in 2° grids. The LAI sample size for each vegetation is listed. (b) The temporal distribution of LAI samples for the eight vegetation biome types and the annual variation of LAI sample size. (c) The distribution of LAI values in percentage (bin width = 0.1) for Landsat LAI samples, GIMMS LAI3g, GLASS LAI, and GLOBMAP LAI. It should be noted that 40,000 Reprocessed MODIS LAI samples were introduced at locations and months when Landsat LAI samples were scarce.

The following changes are made in the revised manuscript:

(a) Inter-annual variation of the Landsat LAI sample size in Section 4.1:
"During 1984−2015, the Landsat LAI sample size per year increased from 28,323 in 1984, peaked at 200,315 in 2001 when both Landsat 5 and Landsat 7 were available, and then leveled off until 2012 (sample size: 22,106) (Figure 2b). From November 2011 to May 2012, very few images were acquired with Landsat 5's decommissioning, (https://www.usgs.gov/centers/eros/science/usgs-eros-archive-landsat-archives-landsat -4-5-thematic-mapper-tm-level-1-data). Since the launch of Landsat 8 in 2013, the Landsat LAI sample became steadily available again." (Page 11, Line 292-297)

**[Comment 15]** *Figure 3 the saturation effect to LAI4g is significant (see my comment at major point #2).*

**[Response 15]** A detailed reply to this comment is referred to **Response 3**. The saturation did exist, and it was more significant for samples whose LAI values deviated from the average (yellow and blue dots in Figure 5). For a majority of samples (red dots in Figure 5), the saturation effect was subtle. The fitting line for GLO in updated Figure 5 is close to the 1:1 line.

**[Comment 16]** *Figure 4 There is data missing (in white color) at southwest China, Euro and southern part of USA.*

**[Response 16]** This study used 2° × 2° grids to evaluate the relative accuracies among GIMMS LAI4g, GIMMS LAI3g, and GLASS LAI. We assumed the evaluation to be reliable only if the LAI sample count within each grid was larger than 100. We have clarified this in the text and Figure 4 caption of the revised manuscript.

The following changes are made in the revised manuscript:

(a) Results interpretation in Section 4.3:

"Grids with a small Landsat LAI sample size (< 100) were excluded as they may not provide a reliable evaluation." (Page 17, Line 378-379)

(b) Figure 6 caption:

"**Figure 6.** Dominance map of the GIMMS LAI3g, GIMMS LAI4g, and GLASS LAI based on their MAE. The map was drawn in 2° × 2° grids whose colors were composed of reciprocal averages of MAE from the GIMMS LAI4g (green), GIMMS LAI3g (red), and GLASS LAI (blue). Non-vegetated grids and grids with small Landsat LAI sample size (< 100) were filled white. A greener grid, for example, indicates that the GIMMS LAI4g has a lower MAE (or a higher absolute LAI accuracy). " (Page 17-18, Line 387-390)

**[Comment 17]** *Figure 6 this is a significant improvement for LAI product based on AVHRR data! Very nice result.*

**[Response 17]** We appreciate it. The removal of effects of NOAA orbital drift and AVHRR sensor degradation is one of the highlights of GIMMS LAI4g.

**[Comment 18]** *Figure 9 authors should add the statistics for each histogram.*

**[Response 18]** Thanks for the suggestion. In the revised manuscript, we have labeled all value frequencies at each LAI interval in the histograms.

[Figure]

**Figure 12.** Global maps of LAI trends and their differences between the global LAI products during 1982−2015. The LAI products include GIMMS LAI4g after consolidation (a), GIMMS LAI3g (b), GLASS LAI (c), and GLOBMAP LAI (d). The trend was calculated as the slope of a linearly fitted LAI time series. (e) to (g) show the slope differences between the GIMMS LAI4g and the other three LAI products.

**[Comment 19]** *Figure 10 authors may add the statistics for the global LAI trend before (pre-MODIS era) and after 2000 (MODIS era). This can provide the insight of whether the vegetated area is greening or browning.*

**[Response 19]** This comment is related to **Comment 4**. Figure 10a (updated Figure 13a) has been augmented to illustrate the vegetation trends before and after 2000.

**[Comment 20]** *L 435 'Two other LAI products, namely the GLASS LAI (1982−2018) and GLOBMAP LAI (1982−2020), also incorporated MODIS data (reflectance). ' This is not totally right, the GLASS-LAI product (1982−2018) is only based on AVHRR data (Xiao et al. 2017).*

*Xiao, Z., Liang, S., & Jiang, B. (2017). Evaluation of four long time-series global leaf area index products. Agricultural and Forest Meteorology, 246, 218-230.*

**[Response 20]** We apologize again for this mistake. We used version 4 of the GLASS LAI product. In this version, GLASS LAI was generated from AVHRR reflectance during 1981−1999 and from MODIS surface reflectance after 2000 (Xiao et al., 2016). Corresponding corrections have been made, as indicated in a previous reply.

Tel: 86 185 0042 6608

Email: zhu.zaichun@pku.edu.cn

**[Comment 1]** *The authors have introduced an updated version of the GIMMS LAI4g datasets, covering the period between 1982 and 2020. BPNN model was trained using PKU GIMMS NDVI, Landsat LAI samples, and three other explanatory variables to predict the LAI values. Also, a pixel-wise linear fusion based method was further used to consolidate the AVHRR-based LAI with the MODIS LAI product. Despite the dataset's design to effectively mitigate the effects of satellite orbital drift and sensor degradation, I believe there is room for further improvement through addressing or discussing the following questions:*

**[Response 1]** We appreciate the reviewer for all the comments. The GIMMS LAI4g product has been generated to mitigate the effect of NOAA satellite orbital drift and AVHRR sensor degradation, as well as to further improve the spatiotemporal consistency and accuracy. We hope our following responses could well address your concerns and the product of GIMMS LAI4g has been further improved.

**[Comment 2]** *Major:*

*Reliability of the samples: two core datasets that used to train the BPNN model, the PKU GIMMS NDVI and Landsat LAI samples are from two unpublished/in-preparation research/articles. This may result in controversy if the authors are unable to provide additional information on these two datasets. a) please provide more details on how the PKU GIMMS NDVI well removed the NOAA orbital drift and AVHRR sensor degradation effects; b) The Landsat LAI samples were trained from MODIS LAI, using what method and sampling strategy? What is the accuracy of the Landsat LAI samples against in-site data?*

**[Response 2]** Thanks for this comment. We have realized that the relatively brief introduction to the PKU GIMMS NDVI product and Landsat LAI samples would raise many concerns about the reliability of these datasets. In the new version of the

manuscript, we have added more details on the generating process of the datasets and their performance in absolute accuracy and vegetation trend.

Specifically, a) we have stated how we used cross-calibrated Landsat NDVI with other explanatory variables to further calibrate GIMMS NDVI3g. The effect of NOAA satellite orbital drift and AVHRR sensor degradation could thus be eliminated in the PKU GIMMS NDVI. Using Evergreen Broadleaf Forests (EBF) as an example, a new figure is now available in the supplementary file (Figure S1) to demonstrate the efficacy of our strategy. b) We have also clarified in the revised manuscript how to generate random sample locations across the globe and how to establish and screen sample pairs between Landsat data and MODIS LAI. The sample pairs were used to build Landsat LAI models which were then applied to the Landsat archive. Landsat LAI samples were eventually created at 40,000 sample locations and were verified against in-situ LAI measurements (BELMANIP and ORNL).

The following changes are made in the revised manuscript:

(a) Introduction to PKU GIMMS NDVI in Section 2.1:

"In the generation of PKU GIMMS NDVI, Landsat NDVI from Thematic Mapper (TM), Enhanced Thematic Mapper Plus (ETM+), and Operational Land Imager (OLI) were first cross-calibrated. Then, massive high-quality Landsat NDVI samples were extracted by considering a series of factors including clouds, cloud shadows, water, snow, aerosol, and radiation performance. The Landsat NDVI samples were employed to calibrate the GIMMS NDVI3g product with other explanatory variables using biome-specific machine learning models. The calibrated NDVI product was finally consolidated with the MODIS NDVI product to extend the temporal coverage to the year 2020. The major improvement of PKU GIMMS NDVI over its counterparts is that it well removed the NOAA orbital drift and AVHRR sensor degradation effects, especially in tropical regions (Figure S1). Its overall $R^2$, Mean Absolute Error (MAE), and Mean Absolute Percentage Error (MAPE) are 0.975, 0.033, and 9%, respectively. It is highly consistent with MODIS NDVI in terms of pixel value ($R^2 = 0.962$, MAE = 0.032, and MAPE = 6.5%) and global vegetation trend. PKU GIMMS NDVI inherited the quality control (QC) information from the GIMMS NDVI3g. A QC value of 0, 1,

and 2 indicates NDVI of good quality, NDVI retrieved from spline interpolation, and NDVI retrieved from average seasonal profile, respectively. The PKU GIMMS NDVI record during AVHRR missions from 1982 to 2015 (before consolidation with MODIS NDVI) was used in this study. It is available at https://zenodo.org/record/7441559#.Y7J7y3ZByCo." (Page 4, Line 97-110)

(b) Introduction to Landsat LAI samples in Section 2.2:

"The Landsat LAI sample dataset provides approximately 4.9 million high-quality samples with a spatial resolution of 1/12 ° and a temporal resolution of half a month (Zha, in preparation). It covers the global vegetated area with all vegetation biome types defined in the MODIS land cover product (the third classification scheme; see section 2.4) and a long-time span from 1984 to 2020. In the generation of Landsat LAI samples, 70,000 sample locations for Deciduous Needleleaf Forests [DNF] and 100,000 sample locations for each of the other vegetation biome types were randomly selected based on the MODIS land cover product. At the sample locations, Reprocessed MODIS LAI (in 500 m resolution; see Section 2.3) and Landsat surface reflectance from TM, ETM+, and OLI scenes (20 × 20 pixels in 30 m resolution) were extracted, creating massive sample pairs. The sample pairs were then rigorously screened by criteria that were not limited to those mentioned in Section 2.1 (i.e., clouds, cloud shadows, etc.) but also included Landsat sample purity, NDVI-LAI relationship, and the saturation state of the MODIS LAI. Biome- and Landsat sensor-specific Random Forest models with other explanatory variables were built based on the sample pairs. The models were applied to historical Landsat data at 40,000 random sample locations (1/12°) to create the final Landsat LAI sample dataset. Validation of the dataset through observations from the BEnchmark Land Multisite ANalysis and Intercomparison of Products (BELMANIP) network (Baret et al., 2006) and the Oak Ridge National Laboratory (ORNL) (Scurlock et al., 2001) showed high absolute accuracies ($R^2$ = 0.76, MAE = 0.45 $m^2/m^2$, Root Mean Square Error (RMSE) = 0.66 $m^2/m^2$). The inter-comparison with the Reprocessed MODIS LAI shows a high temporal consistency. This study selected 3.6 million Landsat LAI samples between 1984 and 2015." (Page 4-5, Line 112-127)

**[Comment 3]** *Model development: As is well known that, NDVI saturates at the dense*

*vegetation areas (LAI around 3), can you explain how the BPNN model utilizes the already saturated NDVI to predict LAI in higher ranges (4-7)?*

**[Response 3]** It is true that NDVI would saturate at dense canopies. To deal with this effect, we introduced other explanatory variables (including longitude, latitude, month, and satellite number and years since launch) and established biome-specific BPNN models to explain the LAI variations in space, time, satellite, and biome. As such, the predicted LAI was not only determined by NDVI values but other explanatory variables. Figure 3 shows the significant contributions from spatial and temporal variables to the BPNN model accuracy. Figure 5 shows that GIMMS LAI4g could be relatively resistant to saturation. The saturation effect for a particular biome was subtle at a majority of sample locations (red dots in Figure 5) and was observable for samples whose LAI values deviated from the average (yellow and blue dots in Figure 5). For sparse vegetation types including GRA, SHR, CRO, and SAV, the GIMMS LAI4 would be underestimated at their high values; and for dense vegetation types including EBF, DBF, ENF, and DNF, the GIMMS LAI4 would be underestimated at their low and medium values. It should be noted that the LAI fitting line for the global vegetation biome is close to the 1:1 line (Figure 5a-9) and the saturation effect can also be observed in EBF, DBF, ENF, and DNF of GLASS LAI, which was not derived from NDVI data but from surface reflectance.

Despite the potential saturation effect, GIMMS-based NDVI products have been among the most reliable and popular ones because they overcome the limitations of the AVHRR instrument or operation, e.g., vicarious calibration, atmospheric and cloud correction, bias correction, satellite orbital drift, and sensor degradation (Pinzon and Tucker, 2014; Li et al., in review). For example, the GIMMS LAI3g (based on GIMMS NDVI3g) is one of the core data references in the IPCC Sixth Assessment Report for the assessment of global vegetation changes (Eyring et al., 2021), and has been cited for more than 600 times (from *Web of Science*).

[Figure]

**Figure 5.** Validation of the (a) GIMMS LAI4g, (b) GIMMS LAI3g, (c) GLASS LAI, and (d) GLOBMAP LAI products in different vegetation biomes using Landsat LAI samples from 1984 to 2015. The error metrics are $R^2$, RMSE, MAE, and MAPE. GLO represents the global vegetation biome. The color of the dots represents LAI value frequencies in a 0.5 ($m^2/m^2$) interval.

The following changes are made in the revised manuscript:

(a) Results on the saturation effect in Section 4.3:

"Compared to Landsat LAI samples, the four global LAI products were underestimated in almost all the vegetation types except for CRO of GLASS LAI (Figure 5). We found certain levels of saturation in GIMMS LAI4g and GIMMS LAI3g, such as for high values of GRA, SHR, CRO, and SAV and medium values of EBF, DBF, ENF, and DNF (Figure 5). This could be attributed to the use of NDVI data in LAI models. However, we also observed the saturation effect in EBF, DBF, ENF, and DNF of GLASS LAI, which was not derived from NDVI data. For GIMMS

LAI4g, the saturation was relatively subtle at a majority of sample locations (red dots) and was obvious for locations whose LAI values deviated from the average (yellow and blue dots). The LAI fitting line of the global vegetation biome in GIMMS LAI4g (Figure 5a-9) was close to the 1:1 line." (Page 17, Line 370-376)

    (b) Discussions on the saturation effect in Section 5.3:

"The use of PKU GIMMS NDVI could also result in the saturation effect in GIMMS LAI4g as NDVI data tend to saturate at high values (Figure 5). This study established biome-specific models that incorporate multiple explanatory variables besides PKU GIMMS NDVI to account for the LAI variations in space, time, biome, and satellite. This effort could help alleviate the saturation effect though the effect still exists." (Page 27, Line 555-558)

**[Comment 4]** *Currently, the validation of the GIMMS LAI4g datasets appears to rely heavily on the Landsat LAI sample data, which has not been published and may not necessarily represent the true LAI values on the ground. As a result, there seems to be a lack of independent validation for the proposed dataset. I would like to suggest that the author use the available field data to validate their dataset.*

**[Response 4]** We have to agree with the reviewer that, the current validation of GIMMS LAI4g based on Landsat LAI samples only could hardly provide solid and comprehensive evidence for its accuracies. In the revised manuscript, independent in-situ LAI data have been introduced. A total of 113 field LAI measurements from BELMANIP 2.1, DIRECT 2.1, and ORNL were employed to evaluate the GIMMS LAI4g product as well as other global long-term LAI products. The results showed that GIMMS LAI4g ($R^2$ = 0.69, RMSE = 0.86 $m^2/m^2$, MAE = 0.61 $m^2/m^2$, MAPE = 33%) had comparable accuracies with GIMMS LAI3g ($R^2$ = 0.71, RMSE = 0.78 $m^2/m^2$, MAE = 0.55 $m^2/m^2$, MAPE = 30%) and GLASS LAI ($R^2$ = 0.68, RMSE = 0.82 $m^2/m^2$, MAE = 0.60 $m^2/m^2$, MAPE = 33%), and it had the lowest underestimation for the fitting line with a slope of 0.87 and an intercept of 0.05 (Figure 4).

[Figure]

**Figure 4.** Validation of the (a) GIMMS LAI4g, (b) GIMMS LAI3g, (c) GLASS LAI, and (d) GLOBMAP LAI products using 113 field LAI measurements from 49 sites in the projects of BELMANIP 2.1, DIRECT 2.1, and ORNL. Sites of different vegetation biome types are marked by colors. The error metrics are $R^2$, RMSE, MAE, and MAPE. The blue fitting lines and dashed 1:1 lines are drawn.

The following changes are made in the revised manuscript:

(a) Introduction to field LAI measurements in Section 2.8:

"The field LAI measurements were from three projects namely, BELMANIP 2.1 (available at https://calvalportal.ceos.org/web/olive/site-description) (Baret et al., 2006), DIRECT 2.1 (available at https://calvalportal.ceos.org/lpv-direct-v2.1) (Garrigues et al., 2008), and ORNL (available at https://daac.ornl.gov/VEGETATION/guides/LAI_guide.html) (Scurlock et al., 2001). The BELMANIP 2.1 and DIRECT 2.1 provide 3 km × 3 km averaged LAI values derived from sites in networks of FLUXNET, AERONET, VALERI, BigFoot, etc. The upscaling from site-based LAI to 3 km × 3 km LAI used high spatial resolution imageries such as Landsat and SPOT. Most global long-term LAI products have utilized the BELMANIP and DIRECT LAI as ground truth for product evaluation (Myneni et al, 2002; Liu et al., 2012; Xiao et al., 2016; Zhu et al., 2013), yet the LAI measurements in both projects were available only after the late 1990s. Note that GLASS LAI (version 4) also employed BELMANIP sites for LAI model training

(Xiao et al., 2016). This study further incorporated ORNL sites which provide field LAI measurements during 1932−2020 despite possible scaling effects due to spatial heterogeneity. We prudently examined all the sites and measurements in BELMANIP 2.1, DIRECT 2.1, and ORNL, and removed measurements that were acquired from heterogeneous sites or coincident among the three projects. Eventually, 113 field LAI measurements from 49 sites were obtained. Information on selected field LAI measurements can be found in Table S1." (Page 6-7, Line 173-187)

(b) Results of validation using field LAI measurements in Section 4.3:
"Based on field LAI measurements, GIMMS LAI4g generated from the BPNN models presented comparable accuracies ($R^2$ = 0.69, RMSE = 0.86 $m^2/m^2$, MAE = 0.61 $m^2/m^2$, MAPE = 33%) with GIMMS LAI3g ($R^2$ = 0.71, RMSE = 0.78 $m^2/m^2$, MAE = 0.55 $m^2/m^2$, MAPE = 30%), and GLASS LAI ($R^2$ = 0.68, RMSE = 0.82 $m^2/m^2$, MAE = 0.60 $m^2/m^2$, MAPE = 33%) (Figure 4). GIMMS LAI3g had the best performance in error measures (i.e., $R^2$, RMSE, MAE, and MAPE), but GIMMS LAI4g had the lowest underestimation for the fitting line with a slope of 0.87 and an intercept of 0.05 (Figure 4). GLOBMAP LAI presented the largest discrepancies from the LAI measurements." (Page 14, Line 335-340)

**[Comment 5]** *Providing global distribution maps of GIMMS LAI4g at representative time points and comparing them with other products may be useful and informative for potential users.*

**[Response 5]** As suggested by the reviewer, we provided LAI distribution maps of multi-year average in January and July for GIMMS LAI4g, GIMMS LAI3g, GLASS LAI, and GLOBMAP LAI (Figure 7).

[Figure]

**Figure 7.** Inter-comparison of spatially averaged LAI along latitude between the GIMMS LAI4g, GIMMS LAI3g, GLASS LAI, and GLOBMAP LAI in January and July of the years 1990, 2000, and 2010. The spatial average was calculated at an interval of 1°.

The following changes are made in the revised manuscript:

(a) GIMMS LAI4g at representative time points in Section 4.3:

"Figure 7 showcases the spatially averaged LAI along latitude in January and July of the years 1990, 2000, and 2010 for GIMMS LAI4g, GIMMS LAI3g, GLASS LAI, and GLOBMAP LAI, respectively. The four LAI products were overall consistent. The GIMMS LAI4g and GIMMS LAI3g had lower values at northern middle latitudes (35°N – 65°N) in July (Figure 7b; Figure 7d; Figure 7f). Also in July, the GLOBMAP LAI and GLASS LAI in the Northern Hemisphere maintained good consistency for the years 1990 and 2010 (Figure 7b; Figure 7f), but the GLOBMAP LAI was systematically lower than GLASS LAI for the year 2000 (Figure 7d). " (Page 18, Line 392-397)

**[Comment 6]** *I suggest that the authors consider including RMSE as an evaluation parameter in their study. RMSE quantifies the variance of errors and is therefore*

*more representative than MAE.*

**[Response 6]** Thank you for this suggestion. We have included RMSE as an error metric in the revised manuscript. Corresponding results including the texts, figures, and tables have been updated. Please check the revised manuscript as this modification was made in substantial locations.

**[Comment 7]** *To provide better clarity on potential under or overestimations for high/low values, please add a 1:1 line to Fig. 3.*

**[Response 7]** This is a good point. The 1:1 lines, LAI fitting lines, and color bars have been added to all sub-figures in Figure 3 (updated Figure 5; see **Response 3**).

**[Comment 8]** *L430 "We chose not to include the vegetation biome type as an explanatory variable (GIMMS LAI3g did) because the values of the vegetation biome type are deterministic rather than continuous", the NOAA satellite number is also deterministic but why it was adopted as the explanatory variable?*

**[Response 8]** The reason is that NOAA missions divided the whole time series (1982−2010s) into periods (Figure 8). Building individual models for different NOAA missions would fail to explain the temporal consistency in LAI. We used NOAA satellite number as an explanatory variable so that the BPNN models provided consistent LAI values from 1982−2010s. This has been clarified in the revised manuscript.

The following changes are made in the revised manuscript:

(a) Discussion on BPNN models in Section 5.1:

"The deterministic NOAA satellite number was used as an explanatory variable so that the temporal consistency of the BPNN models can be ensured." (Page 25, Line 521-522)

Tel: 86 185 0042 6608

Email: zhu.zaichun@pku.edu.cn

**[Comment 1]** *This manuscript describes a new Global Inventory Modeling and Mapping Studies (GIMMS) LAI product (GIMMS LAI4g) for the period 1982-2020. The new product represents a major advance over previous products in that it uses the latest PKU GIMMS NDVI product and a representative subset of Landsat LAI samples to remove sensor degradation and orbital drift that resulted in bias in previous long-term products (e.g., LAI3g). The improved methodology appears to have effectively removed bias and the new product is demonstrated to out-performed other existing widely used LAI products over most of the terrestrial land surface.*

**[Response 1]** We thank the referee for highlighting the significance of this research.

**[Comment 2]** *Overall, I feel the manuscript is well written and the methodology is well-described. I commend the authors for this very important undertaking and the major advance of incorporating Landsat observations to help correct bias due to sensor degradation and orbital drift in the long-term AVHRR record. This is a landmark dataset that will be widely used in advancing our understanding of long-term trends in global vegetation dynamics. I have a few major comments below that need to be addressed before I can recommend acceptance of this manuscript for publication.*

**[Response 2]** We are inspired by this comment. We are happy that the manuscript basically meets the reviewer's standard, and that the reviewer confirms the advantage of eliminating the effect of satellite orbital drift and sensor degradation using the Landsat archive. Efforts have been made to further improve the manuscript and the dataset according to the comments. We hope the revised manuscript could address all the concerns.

**[Comment 3]** *Major Comments:*

*1. The introduction is very well-written and does an excellent job of framing the importance of the research and new product.*

**[Response 3]** Thanks again for this positive comment. Many modifications and improvements have been made in other sections of the manuscript. We hope they are also satisfying.

**[Comment 4]** *Methods 3.1. The PKU GIMMS NDVI spans the 1982-2020 time period. Why don't the authors simple use the BPNN model to predict LAI values for the full record? Why do the authors instead incorporate MODIS LAI after 2003? This seems to add unnecessary sources of uncertainty into the process. It should hold that the PKU GIMMS NDVI is the best available long-term NDVI product and the LAI reference dataset (1984-2020) would be adequate to build a long-term LAI product using the BPNN model alone.*

**[Response 4]** We apologize for the confusion. In the current versions of PKU GIMMS NDVI and GIMMS LAI4g, MODIS products were adopted after 2003 or 2004. AVHRR was the only data source before 2000 that provides spatiotemporally continuous observations, but its availability and data quality have been limited since the late 2010s (https://www.usgs.gov/centers/eros/science/usgs-eros-archive-advanced-very-high-res olution-radiometer-avhrr). For instance, the NDVI data in AVHRR-based GIMMS NDVI3g was available until 2015. From this point, it would be inevitable to consolidate AVHRR-based products with more recent products to acquire up-to-date long-term time series. During the period that two products overlapped, a choice has to be made on which one shall be used. As a more advanced sensor, MODIS has been elaborately verified and is believed to provide more reliable data. As such, in both PKU GIMMS NDVI and GIMMS LAI4g, MODIS data were used during the overlapping period.

With this being said, we agree with the reviewer that it would make lots of sense to provide an AVHRR-based LAI product that is independent of MODIS data. Therefore, we have generated the AVHRR-based LAI product (1982−2015) without

consolidation with Reprocessed MODIS LAI. This new dataset is also available at https://doi.org/10.5281/zenodo.8035760.

The following changes are made in the revised manuscript:

(a) Added information in the Data availability section:

"In the same repository, we have also provided the version of GIMMS LAI4g that is solely based on AVHRR data, which means that its generation was free from the consolidation with Reprocessed MODISL LAI and it used the version of PKU GIMMS NDVI before consolidation with MODIS NDVI." (Page 28, Line 595-597)

**[Comment 5]** *Results 4.1. The LAI reference dataset spans 1984-2020 and relates Landsat reflectance to MODIS LAI using machine learning. This paper describing this dataset is currently in review. I think there should be more detail on this product included in the text. For instance, it needs to be clearly stated which MODIS LAI product was used in the generation of the LAI reference dataset so it's clear that there is consistency with the MODIS LAI used in the data consolidation step.*

**[Response 5]** Thanks for the comment. In the revised manuscript, we have provided more details on the Landsat LAI sample dataset, including its generation process and performance in absolute accuracy and vegetation trend. We hope the updated text could be more informative.

The following changes are made in the revised manuscript:

(a) Introduction to Landsat LAI samples in Section 2.2:

"The Landsat LAI sample dataset provides approximately 4.9 million high-quality samples with a spatial resolution of 1/12 ° and a temporal resolution of half a month (Zha, in preparation). It covers the global vegetated area with all vegetation biome types defined in the MODIS land cover product (the third classification scheme; see section 2.4) and a long-time span from 1984 to 2020. In the generation of Landsat LAI samples, 70,000 sample locations for Deciduous Needleleaf Forests [DNF] and 100,000 sample locations for each of the other vegetation biome types were randomly selected based on the MODIS land cover product. At the sample locations, Reprocessed MODIS LAI (in 500 m resolution; see Section 2.3) and Landsat surface reflectance from TM, ETM+, and OLI scenes (20 × 20 pixels in 30 m resolution) were

extracted, creating massive sample pairs. The sample pairs were then rigorously screened by criteria that were not limited to those mentioned in Section 2.1 (i.e., clouds, cloud shadows, etc.) but also included Landsat sample purity, NDVI-LAI relationship, and the saturation state of the MODIS LAI. Biome- and Landsat sensor-specific Random Forest models with other explanatory variables were built based on the sample pairs. The models were applied to historical Landsat data at 40,000 random sample locations (1/12°) to create the final Landsat LAI sample dataset. Validation of the dataset through observations from the BEnchmark Land Multisite ANalysis and Intercomparison of Products (BELMANIP) network (Baret et al., 2006) and the Oak Ridge National Laboratory (ORNL) (Scurlock et al., 2001) showed high absolute accuracies ($R^2$ = 0.76, MAE = 0.45 $m^2/m^2$, Root Mean Square Error (RMSE) = 0.66 $m^2/m^2$). The inter-comparison with the Reprocessed MODIS LAI shows a high temporal consistency. This study selected 3.6 million Landsat LAI samples between 1984 and 2015." (Page 4-5, Line 112-127)

**[Comment 6]** *Results 4.4. The temporal consistency analysis is critical in demonstrating how effectively sensor degradation and orbital drift bias was removed from the dataset. Currently, only results for EBF are shown (Figure 6). I suggest the authors expand this analysis across biomes and globally. Improved temporal consistency is one of the major advances of this dataset and should be robustly characterized.*

**[Response 6]** We used EBF as an example for two reasons. First, it suffers from a more obvious effect of NOAA satellite orbital drift and AVHRR sensor degradation than other vegetation biomes (Pinzon and Tucker, 2014; Tian et al., 2015; Zhu et al., 2013). Second, EBF has long been a research focus with an abundance of biomass and intensive vegetation activity. In the supplementary materials of the revised manuscript, we have provided results of the temporal consistency analysis for all other vegetation biomes, where different levels of satellite orbital drift and sensor degradation can be effectively eliminated (Figure S4-S10).

**[Comment 7]** *After 2003, the GIMMS LAI4g product directly matches the MODIS*

*BNU LAI product from Yuan et al., (2011). This could create bias in the trend in LAI4g over the AVHRR time period (1982-2003) and over the MODIS time period (2004-2020). Also, given their reliance on different MODIS products, this could introduce differences in the trends of the PKU NDVI product and the LAI4g product. The MODIS BNU LAI dataset used an algorithm that interpolated and smoothed the original MODIS LAI data. For instance, MODIS BNU LAI is impacted by the follow steps used in the algorithm: 1) MODIS LAI retrievals that experienced saturation were flagged as low-quality and discarded; 2) the upper envelope smoothing process may generate positive bias in the trend. It would be helpful for the authors to show the annual anomalies and trend of the various products used in the derivation of LAI4g including PKU NDVI4g (1982-2020), LAI4g no consolidation (1982-2015), LAI4g consolidation (1982-2020), BNU LAI (2003-2020). This could be show for the full global region in the main paper and PFT level trends could be moved to the SI. Please then explain any differences in the PKU NDVI, BNU LAI, and LAI4g trend over the MODIS era (2003-2020).*

*Yuan, H., Dai, Y., Xiao, Z., Ji, D., & Shangguan, W. (2011). Reprocessing the MODIS Leaf Area Index products for land surface and climate modelling. Remote Sensing of Environment, 115(5), 1171–1187. https://doi.org/10.1016/j.rse.2011.01.001*

**[Response 7]** As mentioned in **Response 4**, in order to acquire spatiotemporally consistent LAI products from 1982 to now, it is inevitable to consolidate the irreplaceable AVHRR-based LAI with the LAI product generated from more recent and advanced sensors. This study selected the MODIS BNU LAI (Reprocessed MODIS LAI in the manuscript) in consolidation because it was argued to have higher accuracy due to the spatiotemporal filtering and the use of quality flag (Samanta et al., 2011; Yuan et al., 2011). Note that the MODIS BNU LAI has also served as the LAI target in generating GIMMS LAI3g and our Landsat LAI samples. As a result, the vegetation trend in GIMMS LAI4g is jointly determined by AVHRR data and MODIS data. The bias could hardly be quantified but as the reviewer suggested, we have compared the annual anomalies and trends of GIMMS LAI4g before (1982−2015) and after (1982−2020) consolidation, MODIS BNU LAI (1982−2020), and PKU

GIMMS NDVI4g (1982−2015) (updated Figure 11). We found a good consistency between them. As suggested by the reviewer, the biome-specific version of Figure 11 is available in Figure S12.

[Figure]

**Figure 11.** Annual anomalies and trends of GIMMS LAI4g before consolidation (1982−2015), GIMMS LAI4g after consolidation (1982−2020), Reprocessed MODIS LAI (2004−2020), and PKU GIMMS NDVI (1982−2015).

The following changes are made in the revised manuscript:

(a) Comparison of LAI variations between products in Section 4.4:

"Figure 11 demonstrates a good consistency in the overlapping periods between annual variations of the final GIMMS LAI4g product (GIMMS LAI4g after consolidation) and the input and intermediate products. The shapes of the anomalies were similar. The LAI trends for GIMMS LAI4g remained constant before and after consolidation ($2.9{\times}10^{-3}$ $m^2m^{-2}yr^{-1}$), despite Reprocessed MODIS LAI presenting a high greening trend during 2004−2020. The consistency has also been found in a biome-specific manner (Figure S12). This result indicates that both BPNN modeling (PKU GIMMS NDVI vs GIMMS LAI4g before consolidation) and data consolidation (GIMMS LAI4g before consolidation vs GIMMS LAI4g after consolidation) preserved the LAI anomaly and trend." (Page 22, Line 451-457)

**[Comment 8]** *The GIMMS LAI4g product derived from NDVI4g (before harmonization with MODIS BNU LAI) is a valuable in its own right. This dataset serves as a high-quality estimation of LAI, purely from AVHRR, inter-calibrated with*

*Landsat, and independent of other MODIS-based LAI data. I recommend the authors publish this dataset along with the other final version.*

**[Response 8]** The reply to this comment could be referred to **Response 4**. We agree with the reviewer's opinion and have provided the version of GIMMS LAI4g that is purely based on AVHRR data (1982−2015).

**[Comment 9]** *Minor comments:*

*Figure 5. The LAI4g line should be on top of the others and easier to see.*

**[Response 9]** Thank you for this suggestion. In the Figure 5 (updated Figure 7), the curve related to GIMMS LAI4g has been placed on top of others.

[Figure]

**Figure 7.** Inter-comparison of spatially averaged LAI along latitude between the GIMMS LAI4g, GIMMS LAI3g, GLASS LAI, and GLOBMAP LAI in January and July of the years 1990, 2000, and 2010. The spatial average was calculated at an interval of 1°.

**[Comment 10]** *Figure 7. The figure could be reduced to the global timeseries and a few key regions discussed in the paper. The rest of the timeseries could be moved to*

*the SI.*

**[Response 10]** This is a good suggestion. The new figure (updated Figure 9) has included the GIMMS LAI4g time series at the global scale and in selected regions including Europe (Ciais et al., 2005), Amazon (Wang et al., 2013), Congo (Zhou et al., 2014), China, and India (Chen et al., 2019a). Temporal variations of GIMMS LAI4g for vegetation biome were moved to the Supplementary materials. We found a systematic deviation between the GIMMS LAI4g and MODIS LAI products during 2004−2015 in all regions before data consolidation. The deviation has been efficiently eliminated by our pixel-wise consolidation method, especially for the Amazon rainforests. After consolidation, the temporal variations of the GIMMS LAI4g were self-consistent in all periods. Similar results were found regarding the vegetation biome type (Figure S11).

[Figure]

**Figure 9.** Temporal variations of the GIMMS LAI4g during 1982−2020 in selected hotspot regions of China (a), India (b), Congo (c), Amazon (d), and Europe (e) and at the global scale (f). GLO represents the global vegetation biome. The bold colored line represents the LAI average of GIMMSLAI4g after data consolidation, with shadow covering the value range between 10% and 90% quantiles. The thin black line represents the LAI average of GIMMSLAI4g before consolidation. It should be noted that the GIMMS LAI4g after consolidation shared the same footprint with the Reprocessed MODIS LAI after the year 2004.

The following changes are made in the revised manuscript:

(a) GIMMS LAI4g temporal variation in key regions in Section 4.4:

"Figure 9 shows the GIMMS LAI4g time series before (thin black line) and after (bold colored line) data consolidation from 1982 to 2020 at the global scale and in selected hotspot regions of Europe, Amazon, Congo, India, and China. The GIMMS LAI4g shares the same footprint with the Reprocessed MODIS LAI after the year 2004. Before consolidation, there was a systematic deviation between the GIMMS LAI4g and MODIS LAI products during 2004−2015 in all regions, especially for the Amazon rainforests (Figure 9d). The pixel-wise fusion method has successfully matched the GIMMS LAI4g time series with MODIS LAI, eliminating abnormal shifts in vegetation phenology. This is especially true for the Amazon rainforests, where the phenological curve has been substantially corrected and enhanced by the MODIS LAI (Figure 9d). As a result, the temporal variations of the GIMMS LAI4g after consolidation were self-consistent in all periods. This temporal consistency has also been evaluated regarding the vegetation biome type and similar results were found (Figure S4)." (Page 20, Line 417-425)

---

## Editor Decision (ED1)

**Issues with GIMMS LAI4g in Australia**

Short comment on "Spatiotemporally consistent global dataset of the GIMMS Leaf Area Index (GIMMS LAI4g) from 1982 to 2020" by Cao, S, Li, M, Zhu, Z, Zha, J, Zhao, W, Duanmu, Z, Chen, J, Zheng, Y, Chen, Y. Earth Syst. Sci. Data Discuss. 2023: 1-31. http://doi.org/10.5194/essd-2023-68

**By: Jorge Luis Peña-Arancibia (jorge.penaarancibia@csiro.au) and Zaved Khan (zaved.khan@csiro.au)**

**28/08/2023**

First of all, we would like to commend Cao et al. (2023) for developing a long-term Leaf Area Index (LAI product) which is an important contribution to earth sciences scientific research. While investigating LAI dynamics over the Australian continent at the catchment scale (BoM, 2022), we noticed some unusual LAI values in Alpine catchments (southeast Australia) in the early 1990s. We would like to showcase these issues to Cao et al. (2023), noting that we have not tried to systematically evaluate errors in GIMMS LAI4g and the findings came from our investigations into catchment non-stationarity, for which vegetation plays a key role (Fowler et al., 2022; Gardiya Weligamage et al., 2023). Therefore, we do not investigate this issue in detail and have only scratched the surface about occurrence and speculate about the causes of the issues. We leave the authors to further verify if the issues occur in other continents and other years.

Alpine catchments in Australia generally have snow cover during the austral winter months (June to August). The example in Figure 1 showcases this issue for catchment ID 221201, located in the Alpine area, LAI values during the early 1990s seem unusually high.

[Figure]

**Figure 1 Annual-averaged Leaf Area Index from GIMMS LAI4g (green solid line, and green dashed line showing a 10-year moving average) LAI for catchment ID 221201. Note the high LAI catchment averaged values in the early 1990s.**

The unusually high LAI values, averaged at the annual frequency are widespread and include intermittent lakes, see Figure 2 for 1992.

[Figure]

**Figure 2 Annual-averaged pixel LAI for southeast Australia, the colour ramp units are in m²/m².**

We further assessed this by obtaining average continental values during August (when snow cover is generally at its largest extent) of 1991 and 1992, and masking all pixels with LAI values >50, see Figure 3. Besides Alpine areas with snow cover, the unusually high values are evident in intermittent dry lakes as well.

[Figure]

**Figure 3 August-averaged pixel LAI for Australia, LAI values above 50 are shown in in yellow. Red dot in the southeast denotes a pixel used to assess the entire timeseries.**

We picked one yellow-masked pixel in the Alpine area to assess this issue in years other than 1991 and 1992. Figure 4 suggest that the issue is not present in other years.

[Figure]

**Figure 4 Monthly pixel LAI for red pixel in Figure 3, Y-axis units are in m²/m².**

The limited investigation of high LAI values in 1991 and 1992 suggest that this is limited to landscapes with high reflectance (snow, dry lake beds) and that filtering these values as for other years is a suitable solution.

**References**

BoM, 2022. Hydrologic Reference Stations, Bureau of Meteorology. http://www.bom.gov.au/water/hrs/, last access: July 2022.

Cao, S., Li, M., Zhu, Z., Zha, J., Zhao, W., Duanmu, Z., Chen, J., Zheng, Y., Chen, Y., 2023. Spatiotemporally consistent global dataset of the GIMMS Leaf Area Index (GIMMS LAI4g) from 1982 to 2020. Earth Syst. Sci. Data Discuss., 2023: 1-31. http://doi.org/10.5194/essd-2023-68

Fowler, K. et al., 2022. Explaining changes in rainfall–runoff relationships during and after Australia's Millennium Drought: a community perspective. Hydrol. Earth Syst. Sci., 26(23): 6073-6120. http://doi.org/10.5194/hess-26-6073-2022

Gardiya Weligamage, H., Fowler, K., Peterson, T.J., Saft, M., Peel, M.C., Ryu, D., 2023. Partitioning of Precipitation Into Terrestrial Water Balance Components Under a Drying Climate. Water Resources Research, 59(5): e2022WR033538. http://doi.org/https://doi.org/10.1029/2022WR033538

---

## Author Response (AR2)

**Reply to Reviewers**

Dear reviewers,

We would like to sincerely express our gratitude again for you taking the time and efforts to review our manuscript entitled "**Spatiotemporally consistent global dataset of the GIMMS Leaf Area Index (GIMMS LAI4g) from 1982 to 2020**" (essd-2023-68) submitted to *Earth System Science Data*. We have carefully considered all the comments and incorporated most of the suggestions, which have further improved the manuscript. We hope that the new version of the manuscript has addressed all the concerns.

Below we provide point-to-point responses, each following the specific comment from the reviewer. All the changes in the revised manuscript have been marked in red.

Sincerely yours,

Zaichun Zhu, Ph. D. (on behalf of the author team)

School of Urban Planning and Design

Peking University

Tel: 86 185 0042 6608

Email: zhu.zaichun@pku.edu.cn

**Referee 1**

**[Comment 1]** *I appreciate the authors adding new and supportive results to the new manuscript. The current version of the manuscript and the dataset has been substantially improved. There are only a few minor points for the authors to improve their manuscript.*

**[Response 1]** We thank all the constructive comments from the reviewer. We hope the minor points have been well addressed in this updated version of the manuscript.

**[Comment 2]** *1. The method used the consolidation with reprocessed MODIS LAI at the MODIS-era, so the trend after 2000 should be highlighted that it is depended on the trend from MODIS. The LAI trend after 2000 is not only from Landsat and AVHRR but also MODIS, which is an important point that should be pointed out in the abstract.*

**[Response 2]** Thanks for this suggestion. In the Abstract of the updated manuscript, we have now stated that "The consolidation with the reprocessed MODIS LAI allows the GIMMS LAI4g to extend the temporal coverage from 2015 to recent (2020), producing the LAI trend that maintains high consistency before and after 2000 and aligns with the MODIS LAI trend during the MODIS era." (Page 2, Lines 30-32).

**[Comment 3]** *2. I would also thank the author pay attention to the LAI trend analysis between pre-MODIS era and MODIS era. However, I think the results may not right. Particularly, the EBF could not have a decreasing trend at the pre-MODIS era. This may because of the AVHRR or Landsat data have limitation at the first a few years of observation. Moreover, for the trend analysis at the MODIS era, I recommend using the LAI data after 2003. The authors may refer to Yuan et al. (2011). So for the LAI trend analysis at the pre-MODIS era and MODIS era, it should be corresponded to 1984-1999 and 2003-2020, respectively.*

*Yuan, H., Dai, Y., Xiao, Z., Ji, D., & Shangguan, W. (2011). Reprocessing the MODIS Leaf Area Index products for land surface and climate modelling. Remote Sensing of Environment, 115(5), 1171-1187.*

**[Response 3]** We admire the reviewer's incisive comment on the decreasing trend of EBF in the pre-MODIS era (before 2000) for GIMMS LAI4g. We find that the reason, as pointed out by the reviewer, is a combination of limitations in the AVHRR and Landsat data. The GIMMS LAI4g was based on the AVHRR/NOAA-7 data from January 1982 to February 1985. Meanwhile, the Landsat LAI samples have been absent before 1984 and were scarce in 1984 (3500 per biome on average and 620 for EBF) (Figure 2b). To address this issue, we have applied the version of BNPP models with explanatory variables of PKU GIMMS NDVI, spatial information, and temporal information only (i.e., NOAA satellite number and years since launch excluded) on the period of 1982−1984 for all vegetation biomes. It should be noted that, this version of BNPP models ($R^2$: 0.95; RMSE: 0.46 $m^2/m^2$; MAE: 0.27 $m^2/m^2$; MAPE: 12.27%) only showed slightly lower accuracies than the one used before ($R^2$: 0.95; RMSE: 0.45 $m^2/m^2$; MAE: 0.27 $m^2/m^2$; MAPE: 11.98%) (Figure 3). The abnormal decreasing trend of EBF in the pre-MODIS era has now been amended (Figure 13).

On the other hand, the division of pre- and post-MODIS era in this study was determined not only by the MODIS data used in GIMMS LAI4g (the Reprocessed MODIS LAI; 2004−2020), but also by those used in other LAI products. For instance, both GLASS LAI (version 4) and GLOBMAP LAI (version 3) used MODIS surface reflectance since 2000. To better compare the temporal consistency before and after the MODIS era for different LAI products, we have to compromise between the MODIS data and finally chose 2000 as the dividing year. We hope the reviewer understands our rationale.

[Figure]

**Figure 13.** Variations of annual LAI anomaly of different vegetation biomes in the global LAI products during 1982−2015. The LAI products include GIMMS LAI4g, GIMMS LAI3g, GLASS LAI, and GLOBMAP LAI. (a) shows the slope values of the annual LAI during 1982−2015 (p1), 1982−2000 (p2), and 2001−2015 (p3). In the x-axis, 4g, 3g, GLA, and GLO stands for GIMMS LAI4g, GIMMS LAI3g, GLASS LAI, and GLOBMAP LAI, respectively. (b) shows the annual LAI time series.

The following changes are made in the revised manuscript:

● Results in determining the optimum BPNN models in Section 4.2:
"As such, for most periods during 1982−2015, the BPNN models adopted the combination of all explanatory variables (S5), including NDVI, longitude, latitude, month, NOAA satellite number, and NOAA satellite in orbit duration. For the period of 1982−1984, the BPNN models adopted the combination of NDVI, longitude, latitude, and month (S3) because of the acceptable accuracies (Figure 3 and Table 1) and the absence (before 1984)

and scarce (1984) Landsat LAI samples (see section 4.1) which could lead to a biased derivation of LAI." (Page 15, Lines 346-350)

• Discussions on the uncertainty source of GIMMS LAI4g product in Section 5.3:

"Also, the Landsat LAI samples were absent before 1984 and scarce in 1984 (section 4.1), which would produce larger uncertainties for the GIMMS LAI4g product during the NOAA-7 period (July 1981 to February 1985)." (Page 29, Lines 580-582)

**[Comment 4]** *Besides, there are two types of EBF, one is in tropical and the other one is in temperate region. To my humble knowledge, I think the tropical EBF could not have high LAI increasing rate because of the VPD stress (Yuan et al. 2019). Thus, the authors may also analysis the LAI trend at Amazon and Congo basin as the typical LAI trend of EBF at tropical region.*

*Yuan, W., Zheng, Y., Piao, S., Ciais, P., Lombardozzi, D., Wang, Y., ... & Yang, S. (2019). Increased atmospheric vapor pressure deficit reduces global vegetation growth. Science advances, 5(8), eaax1396.*

**[Response 4]** We thank the reviewer for raising this point, which helps us further improve our data in the important EBF ecosystems. Similar to **Comment 3/Response 3**, we also find limited Landsat LAI samples in EBF from October to April, especially for the years 2012 and 2013 (Figure 2b; Section 4.1). During this period, the scarce Landsat LAI samples had led to an overestimated greening trend for GIMMS LAI4g after 2000. Therefore, in EBF we have also adopted the version of BNPP models without explanatory variables of NOAA satellite number and years since launch, for all periods of October to April during 1982−2015. The high LAI increasing rate in EBF of GIMMS LAI4g has disappeared, and the trend is more reasonably consistent compared to other products (Figure 13).

To further investigate the LAI trends at Amazon and Congo basins, we have analyzed the LAI anomalies using GIMMS LAI4g before consolidation (1982−2015), GIMMS LAI4g after consolidation (1982−2020), GIMMS LAI3g (1982−2015), GLASS LAI (1982−2015), and GLOBMAP LAI (1982−2015), as shown in Figure S16.

[Figure]

**Figure S16.** Annual anomalies ($m^2 m^{-2}$) and trends of GIMMS LAI4g before consolidation (1982−2015), GIMMS LAI4g after consolidation (1982−2020), GIMMS LAI3g (1982−2015), GLASS LAI (1982−2015), and GLOBMAP LAI (1982−2015) in the Congo (a) and Amazon (b) forests.

All the products presented a greening trend in the Congo forests except the GIMMS LAI3g ($-4.7 \times 10^{-3}$ $m^2 m^{-2} yr^{-1}$), with GLASS LAI ($6.1 \times 10^{-3}$ $m^2 m^{-2} yr^{-1}$) presenting the largest slope, followed by GIMMS LAI4g (before consolidation: $3.4 \times 10^{-3}$ $m^2 m^{-2} yr^{-1}$; after consolidation: $3.0 \times 10^{-3}$ $m^2 m^{-2} yr^{-1}$) and GLOBMAP LAI ($2.6 \times 10^{-3}$ $m^2 m^{-2} yr^{-1}$). In the Amazon forests, the GIMM LAI4g before ($1.4 \times 10^{-3}$ $m^2 m^{-2} yr^{-1}$) and after consolidation ($1.9 \times 10^{-3}$ $m^2 m^{-2} yr^{-1}$), GLASS LAI ($6.9 \times 10^{-3}$ $m^2 m^{-2} yr^{-1}$), and GIMMS LAI3g ($1.9 \times 10^{-3}$ $m^2 m^{-2} yr^{-1}$), had a greening trend while the GLOMAP LAI had a browning trend ($-1.8 \times 10^{-3}$ $m^2 m^{-2} yr^{-1}$). Therefore, in the updated version of GIMMS LAI4g, both the Congo and Amazon basins showed moderate but persistent green trends.

The following changes are made in the revised manuscript:

• Results in determining the optimum BPNN models in Section 4.2:

"Similarly, S3 was also adopted in the winter of ENF (Northern Hemisphere: October to April; Southern Hemisphere: May to September) and October to April of EBF due to the limited Landsat LAI samples (Figure 2)." (Page 15, Lines 351-352)

• Results of LAI trend analysis in Section 4.5:

"We also paid attention to the vegetation trends in the EBF of Amazon and Congo (Figure S16). Large inconsistencies were found between the LAI products. Almost all the LAI products presented a greening trend except the GIMMS LAI3g in the Congo forests (-4.7×10$^{-3}$ $m^2m^{-2}yr^{-1}$) and the GLOMAP LAI in the Amazon forests (-1.8×10$^{-3}$ $m^2m^{-2}yr^{-1}$). The GIMMS LAI4g had moderate greening trends compared to other products." (Page 26, Lines 510-513)

**[Comment 4]** *3. The other point I want to point out for the LAI trend is, since figure 13 analyzes the LAI trend among PFT. It is not clear whether the PFT fraction will change from 1982-2020. Thus, I think the LAI trend among PFT only based on the pixels where PFT fraction is unchanged among the investigated years. For example, the authors may use the PFT fraction with unchanged pixels from MCD12Q1 land cover map. The LUH2 is also acceptable for getting the PFT unchanged pixels.*

*Hurtt, G. C., Chini, L., Sahajpal, R., Frolking, S., Bodirsky, B. L., Calvin, K., ... & Zhang, X. (2020). Harmonization of global land use change and management for the period 850–2100 (LUH2) for CMIP6. Geoscientific Model Development, 13(11), 5425-5464.*

**[Response 4]** In this study, the land cover map was derived from the MCD12Q1 product, with the type of each pixel determined as the most frequent PFT during 2001−2019. "The most frequent PFT" represents a balance between the sample size and sample quality for GIMMS LAI4g generation and validation. We decided not to employ unchanged pixels only, as they were insufficient in quantity and spatially unrepresentative. However, we agreed that a trend analysis based on unchanged pixels would be beneficial in the future to reveal the vegetation dynamics in those relatively undisturbed regions, despite being out of the scope of the current study.

● Discussions on the uncertainty source of GIMMS LAI4g product in Section 5.3:

"In addition, this study used a static global land cover map determined by the most frequent biome type within each grid between 2001 and 2019. This strategy could bring potential uncertainties yet represents a balance between the sample size and sample quality for GIMMS LAI4g generation and validation. When applying GIMMS LAI4g for vegetation trends analysis, a careful consideration of land cover change is suggested." (Page 29, Lines 589-592)

**Referee 3**

**[Comment 1]** *The authors have provided detailed revisions and explanations; however, there are still two major questions remaining:*

**[Response 1]** We apologize for the insufficient information provided in the last version of the manuscript, especially those related to the PKU GIMMS NDVI and Landsat LAI samples. We thank the reviewer for raising these issues again. In the following responses, we have tried our very best to elaborate as many details as we can. We hope all the issues have been addressed in the updated manuscript.

**[Comment 2]** *1. More information is needed regarding the PKU GIMMS NDVI (Li et al., in review) used for algorithm development: a) How was cross-calibration implemented for TM, ETM+, and OLI NDVI? b) Regarding the extraction of "massive high-quality Landsat NDVI samples" considering factors such as clouds, cloud shadows, water, snow, aerosol, and radiation performance, do you mean that these samples were screened out for the presence of these factors? c) Please provide more details on the "other explanatory variables" mentioned.*

**[Response 2]** We thank the reviewer for the detailed suggestions regarding the clarification of our method. Following the reviewer's suggestion, the following details have been complemented in the revised manuscript subject to the concerns (a), (b), and (c). These details are also available in Li et al. (in press) which has been recently accepted for publication in ESSD (https://doi.org/10.5194/essd-2023-1).

(a) In the cross-calibration between TM, ETM+, and OLI NDVI, we adopted the method from Berner et al. (2020) to calibrate TM and OLI NDVI to the ETM+ level via BPNN. 100,000 sample locations were randomly selected for each vegetation biome type from the MCD12Q1 (500 m resolution). 20 × 20 Landsat pixels (30 m resolution) were extracted at each sample center from TM, ETM+, and OLI images during the overlapping periods. The sample locations were then refined by removing those (1) with a high atmospheric opacity (information provided by Landsat products) and (2) where more than 10% of the 20 × 20 Landsat pixels have low quality. The low quality was determined by the cloud coverage in the associated Landsat scenes (> 80%), the clouds, cloud shadows, water, or snow judged by the CF Mask algorithm (Zhu et al., 2015), and implausibly high (>1) or extremely low (0.001) surface reflectance. NDVI was calculated and averaged from high-quality pixels at the remaining sample locations. The sample locations were divided into 80% for BPNN training and 20% for BPNN evaluation. In the BPNN model, the explanatory variables included the NDVI of TM or OLI, the image acquisition time (day of the year), and the sample location's spatial coordinates (longitude and latitude). The target variable was the NDVI of ETM+.

(b) In the Landsat sample selection for generating the PKU GIMMS NDVI, we screened out samples that suffered from Mount Pinatubo eruption (August 1991 to December 1992), a high atmospheric opacity (information provided by Landsat products), and low quality. The quality of the sample was determined in the same way as in Landsat NDVI cross-calibration, i.e., a sample was considered as low-quality if most of the Landsat pixels were affected by clouds, cloud shadows, water, snow, aerosol, and implausible radiation performance.

(c) In training the BPNN model for PKU GIMMS NDVI, the explanatory variables for a particular sample include (1) its GIMMS NDVI3g value, (2) the longitude and latitude of the location, (3) the month that the sample belongs, (4) the NOAA satellite number that the GIMMS NDVI3g data was acquired from, and (5) years since the NOAA satellite launched. To this end, the "other explanatory variables" are the longitude and latitude, the month, and the NOAA satellite number and years since its launch.

In the updated manuscript, all the mandatory information of the abovementioned details has been added. However, we also compromise a few details to avoid possible repetition with the original paper (Li et al., 2023).

The following changes are made in the revised manuscript:

● Introduction to PKU GIMMS NDVI in Section 2.1:

"In the generation of PKU GIMMS NDVI, Landsat NDVI from Thematic Mapper (TM), Enhanced Thematic Mapper Plus (ETM+), and Operational Land Imager (OLI) were first cross-calibrated by adjusting the TM and OLI NDVI to the ETM+ level via random sample locations and the BPNN model (Berner et al., 2020). The sample locations were refined by removing those with high atmospheric opacity and low quality which was defined by the occurrence of clouds, cloud shadows, water, or snow and implausible radiation performance. In the BPNN model, the explanatory variables included the NDVI of TM or OLI, the image acquisition day of the year, and the sample location's longitude and latitude; and the target variable was the NDVI of ETM+.

After cross-calibration, massive high-quality Landsat NDVI samples were extracted by screening out samples that suffered from the Mount Pinatubo eruption (August 1991 to December 1992) as well a high atmospheric opacity and a bad quality (same as sample screening in cross-calibration). The Landsat NDVI samples were employed to calibrate the GIMMS NDVI3g product with other explanatory variables (the longitude and latitude, NDVI month, and NOAA satellite number and years since its launch) using biome-specific machine learning models." (Page 4, Lines 105-116)

**[Comment 3]** *The same issue arises with Landsat LAI samples. In line 120, "Biome- and Landsat sensor-specific Random Forest models with other explanatory variables were built based on the sample pairs." Please indicate what the explanatory variables are.*

**[Response 3]** In training the Random Forest model for the Landsat LAI sample dataset, the explanatory variables included Landsat surface reflectance, spectral indices (NDVI, NDWI, and EVI), the longitude and latitude of the sample, and solar zenith and azimuth

angles at the sample location. Therefore, the "other explanatory variables" are spectral indices, the longitude and latitude, and the solar zenith and azimuth angles.

The following changes are made in the revised manuscript:

● Introduction to Landsat LAI sample dataset in Section 2.2:
"Biome- and Landsat sensor-specific Random Forest models with other explanatory variables (NDVI, Normalized Difference Water Index [NDWI], Enhanced Vegetation Index [EVI], the longitude and latitude, and the solar zenith and azimuth angles) were built based on the sample pairs." (Page 5, Lines 136-138)

**[Comment 4]** *2. Concerning the Landsat LAI samples used for model training and validation: a) The distributions of Landsat LAI samples in lower values (0-1) account for a very high percentage (Fig.2c), which does not guarantee "good representativeness". Please provide an explanation for this. b) Are the Landsat LAI samples used for validation and consistency evaluation independent in terms of geolocation and temporal coverage? Please clarify.*

**[Response 4]** For concern (a), it may look counter-intuitive that a majority of Landsat LAI samples were distributed in lower values of 0-1. However, this value distribution agreed well with existing global long-term LAI products (GIMMS LAI3g, GLASS LAI, and GLOBMAP LAI) where statistics were conducted based on all terrestrial vegetation pixels (Figure 2c). This could be further confirmed by the work of Ma and Liang (2022). When developing the GLASS 250-m LAI, they presented a similar LAI distribution pattern via their Figure 4c as below. The dominant low LAI values could be explained by a large portion of the low-LAI vegetation biomes on Earth (grassland, shrubland, and savanna), the non-growing seasons for vegetation biomes such as cropland and deciduous forests, and the snow cover in the winter.

[Figure]

**Fig. 4.** Distribution of LAI values (bin width is 0.1) of the 2014–2015 time-series fused LAI samples at the representative pixels, and MODIS, PROBA-V, and GLASS V5 LAI at global land pixels: (a) "max LAI" represents the distribution of the maximum LAI values of the 2014–2015 LAI time-series, (b) "mean LAI" represents the distribution of the mean LAI values of the 2014–2015 LAI time-series, and (c) "LAI" represents the distribution of all the LAI values of the 2014–2015 time series; and distribution of (d) solar zenith, (e) view zenith and (f) relative azimuth angles (bin width is 1°) of the 2014–2015 time-series MODIS surface reflectance data at the representative pixels and the global land pixels.

For concern (b), we randomly divided the Landsat LAI sample dataset into 80% for BPNN model training and 20% for LAI product evaluation (the second paragraph in Section 3.1). Thus, these two groups of samples were independent of each other in terms of geolocation and temporal coverage. In this study, the evaluation group of Landsat LAI samples (20%) was used for both validation and consistency evaluation.

To avoid confusion, the following changes are made in the revised manuscript:

● In Figure 2 caption:

"For GIMMS LAI3g, GLASS LAI, and GLOBMAP LAI, the value distribution was calculated based on all terrestrial vegetation pixels." (Page 12, Lines 327-328)

**[Comment 5]** *Minor comments:*

*1. In line 210, please describe how the MODIS LAI samples were selected.*

**[Response 5]** When training the BPNN model, we found that the Landsat LAI sample size could be limited in certain locations and months (Figure 2a and Figure 2b). This is especially true for the northern mid-high latitudes in the winter due to the polar night

phenomenon and low solar altitude angle, which means that low LAI values have been lacking in the Landsat LAI samples for some vegetation biome types (i.e., grassland, shrubland, savanna, and ENF; see Figure 2a and Figure S3). To make the biome-specific BPNN models more accountable, we randomly selected 40,000 MODIS LAI samples at latitudes > 25° N in the winter months (October to April), 10,000 for each of the four vegetation biomes (grassland, shrubland, savanna, and ENF). It should be noted that evergreen broadleaf forests in the tropics did not suffer from the LAI values bias due to their consistently high LAI values across the year.

The following changes are made in the revised manuscript:

• BPNN model establishment in the second paragraph of section 3.1:
"Specifically, 10,000 Reprocessed MODIS LAI values were randomly introduced for each of GRA, SHR, SAV, and ENF at latitudes > 25° N in the winter months (October to April)." (Page 9, Lines 230-231)

**[Comment 6]** *2. In Figure 11, please explain why the trend of GIMMS LAI4g is different from the MODIS LAI in slopes after consolidation. Readers may be confused, as the GIMMS LAI4g has already been matched with the MODIS LAI product using the pixel-wise fusion method.*

**[Response 6]** Thanks for this comment. In Figure 11, the trend of GIMMS LAI4g after consolidation and that of MODIS LAI were calculated based on different periods. For GIMMS LAI4g after consolidation, the period was 1982-2020; and for the MODIS LAI, the period was 2004−2020. They were thus not comparable. We apologize for the confusion in the figure, which has been clarified in the updated version of the manuscript.

In fact, the slope of GIMMS LAI4g after consolidation during 2004−2020 had the same value (0.0056) as the MODIS LAI. In Figure 11, a very similar LAI anomaly pattern during 2004−2020 could be found between GIMMS LAI4g after consolidation and the Reprocessed MODIS LAI.

The following changes are made in the revised manuscript:

● In Figure 11 caption:

"Note that the regression equations within the square brackets were calculated from different periods depending on the products." (Page 24, Lines 486-487)

**[Comment 7]** *3. The previous comment on "Providing global distribution maps of GIMMS LAI4g at representative time points and comparing them with other products may be useful and informative for potential users" has not been addressed. I suggest adding the global LAI distribution maps, similar to the global land cover map in Figure S3.*

**[Response 7]** Thank you for raising this issue again. We sincerely apologize for this omission. In the previous version of the manuscript, only LAI variation curves at representative time points were added (Figure 7). We have now provided global distribution maps of GIMMS LAI4g in January and July of the years 1990 (Figure S4), 2000 (Figure S5), and 2010 (Figure S6), with a similar form as Figure S3 in the updated supplementary file. We hope this issue has been addressed.

[Figure]

**Figure S4.** Illustrations of global distribution maps of GIMMS LAI4g after consolidation (a and b), GIMMS LAI3g (c and d), GLASS LAI (e and f), and GLOBMAP LAI (g and h) in January and July of **1990**.

[Figure]

**Figure S5.** Illustrations of global distribution maps of GIMMS LAI4g after consolidation (a and b), GIMMS LAI3g (c and d), GLASS LAI (e and f), and GLOBMAP LAI (g and h) in January and July of **2000**.

[Figure]

**Figure S6.** Illustrations of global distribution maps of GIMMS LAI4g after consolidation (a and b), GIMMS LAI3g (c and d), GLASS LAI (e and f), and GLOBMAP LAI (g and h) in January and July of **2010**.

The following changes are made in the revised manuscript:

• GIMMS LAI4g at representative time points in Section 4.3:

"The global distribution maps of LAI in January and July can be found in Figure S4–S6." (Page 20, Lines 422-423)

**[Comment 8]** *4. DIRECT 2.1 provides more field data and sites than this study used. Please add the quality control method for selecting "113 field LAI measurements from 49 sites".*

**[Response 8]** It is true that, DIRECT 2.1, together with BELMANIP 2.1 and ORNL, has much more field LAI measurements than those used in this study. For instance, DIRECT 2.1 presents 276 measurements from 172 sites. However, in terms of a spatial resolution of 1/12° and a temporal resolution of half-moth, the sites could be spatially (e.g., between Hailun_B and Hailun_C) and temporally overlapped (e.g., measurements on June 19th and 24th of 2012 at Honghe_D). In these cases, we averaged the measurements falling in the same spatial or temporal domain. On the other hand, BELMANIP 2.1 and DIRECT 2.1 provide 3 km × 3 km averaged LAI values and ORNL is site-based. The spatial mismatch between the global long-term LAI products and the field measurements could introduce uncertainties associated with geo-location errors and the scaling effect (Zhu et al., 2013). Thus, we carefully examined all the field LAI measurements and excluded those located in a heterogeneous landscape (i.e., different vegetation biome types) within an 8 km × 8 km window (approximately 1/12°).

The following changes are made in the revised manuscript:

• Introduction to field LAI measurements in section 2.8:
"We prudently examined all the measurements in BELMANIP 2.1, DIRECT 2.1, and ORNL, and removed those that were acquired from heterogeneous sites using an 8 km × 8 km window (approximately 1/12°). Redundant measurements among the three projects were also removed. In a spatial resolution of 1/12° and a temporal resolution of half-moth, we averaged the measurements falling in the same spatial or temporal domain." (Page 7, Lines 200-203)

**References:**

Berner, L. T., Massey, R., Jantz, P., Forbes, B. C., Macias-Fauria, M., Myers-Smith, I., Kumpula, T., Gauthier, G., Andreu-Hayles, L., Gaglioti, B. V., Burns, P., Zetterberg, P., D'Arrigo, R., and Goetz, S. J.: Summer warming explains widespread but not uniform greening in the Arctic tundra biome, Nat Commun, 11, 1-12, https://doi.org/10.1038/s41467-020-18479-5, 2020.

Ma, H. and Liang, S.: Development of the GLASS 250-m leaf area index product (version 6) from MODIS data using the bidirectional LSTM deep learning model, Remote Sens. Environ., 273, 112985, https://doi.org/10.1016/j.rse.2022.112985, 2022.

Zhu, Z., Bi, J., Pan, Y., Ganguly, S., Anav, A., Xu, L., Samanta, A., Piao, S., Nemani, R. R., and Myneni, R. B.: Global data sets of vegetation Leaf Area Index (LAI)3g and Fraction of Photosynthetically Active Radiation (FPAR)3g derived from Global Inventory Modeling and Mapping Studies (GIMMS) Normalized Difference Vegetation Index (NDVI3g) for the period 1981 to 2011, Remote Sens., 5, 927-948, https://doi.org/10.3390/rs5020927, 2013.

Zhu, Z., Wang, S. X., and Woodcock, C. E.: Improvement and expansion of the Fmask algorithm: cloud, cloud shadow, and snow detection for Landsats 4-7, 8, and Sentinel 2 images, Remote Sens Environ, 159, 269-277, https://doi.org/10.1016/j.rse.2014.12.014, 2015.

---

## Author Response (AR3)

**Reply to the issue in Australia**

Dear Dr. Jorge Luis Peña-Arancibia, Dr. Zaved Khan, and Dr. Dalei Hao,

We would like first to express our sincere gratitude for your interest in the GIMMS LAI4g product and your kind report on the issue in Australia. Based on the report, we have carefully inspected the issue and found that the causation could be the invalid data value (fill value: 65535) in the dataset. We recommend properly handling the value in the analysis. We hope our clarification could address the concern and enhance the confidence in GIMMS LAI4g from the community. In the meantime, we enthusiastically welcome future feedback from researchers of all backgrounds.

Although the clarification on the issue does not affect the latest version of GIMMS LAI4g (v1.2), we do make modifications to our manuscript entitled "Spatiotemporally consistent global dataset of the GIMMS Leaf Area Index (GIMMS LAI4g) from 1982 to 2020" (essd-2023-68). Below, we reply to the report as a regular reply letter. Changes in the revised manuscript have been marked in red.

Sincerely yours,

Zaichun Zhu, Ph. D. (on behalf of the author team)

School of Urban Planning and Design

Peking University

Tel: 86 185 0042 6608

Email: zhu.zaichun@pku.edu.cn

**[Comment 1]** *First of all, we would like to commend Cao et al. (2023) for developing a long-term Leaf Area Index (LAI product) which is an important contribution to earth sciences scientific research. While investigating LAI dynamics over the Australian continent at the catchment scale (BoM, 2022), we noticed some unusual LAI values in Alpine catchments (southeast Australia) in the early 1990s. We would like to showcase these issues to Cao et al. (2023), noting that we have not tried to systematically evaluate errors in GIMMS LAI4g and the findings came from our investigations into catchment non-stationarity, for which vegetation plays a key role (Fowler et al., 2022; Gardiya Weligamage et al., 2023). Therefore, we do not investigate this issue in detail and have only scratched the surface about occurrence and speculate about the causes of the issues. We leave the authors to further verify if the issues occur in other continents and other years.*

**[Response 1]** We thank the research team in CSIRO again for providing a detailed report, which excellently describes the issue and the concerns on the GIMMS LAI4g product. The report has explicitly guided us to examine the data in Australia (and across the globe) and to repeat the issue that the team has met. In the following response, we will point out that, **the causation of the issue is likely the inclusion of invalid or fill value of 65535 in the analysis.** The fill value is assigned when the algorithm fails to provide a biophysically meaningful estimate (e.g., negative LAI) which generally occurs in non-vegetation pixels such as snow-cover regions or intermittent lakes. The solution to this issue could be using the latest version of GIMMS LAI4g and properly handling the fill value in the analysis. The use of 65535 as the fill value has been stated in the data repository as well as the Readme file. However, we now include it in the manuscript.

The following changes are made in the revised manuscript:

• Data availability:
"Before applying the GIMMS LAI4g product, we highly recommend users to read the Readme file in the repository and to properly handle the fill value and the quality control flag in the dataset." (Page 31, Lines 630-632)
• Acknowledgments:

"We also thank Dr. Jorge Luis Peña-Arancibia and Dr. Zaved Khan from CSIRO for bringing up issues in southeast Australia during the preprint posting of the manuscript." (Page 31, Line 649-650)

[Comment 2] *Alpine catchments in Australia generally have snow cover during the austral winter months (June to August). The example in Figure 1 showcases this issue for catchment ID 221201, located in the Alpine area, LAI values during the early 1990s seem unusually high.*

[Figure]

Figure 1 Annual-averaged Leaf Area Index from GIMMS LAI4g (green solid line, and green dashed line showing a 10-year moving average) LAI for catchment ID 221201. Note the high LAI catchment averaged values in the early 1990s.

[Response 2] As the BoM (2022) only provides a point location (149.200°E, 37.373°S) of Catchment 221201, we first manually delineate the boundary based on the map (Figure R1).

[Figure]

**Figure R1.** The location of Catchment 221201. The boundary of the catchment is manually delineated by the coordinate (149.200°E, 37.373°S) provided by BoM (2022), as seen in the subfigure of the left-bottom corner.

Despite the possible spatial mismatch, our delineation basically repeats the unusually high GIMMS LAI values for 1991 and 1992 in product version 1.0 (Figure R2a, the black line). However, when the fill value (65535) is ignored in the averaging operation, unusually high LAI values disappear (Figure R2a, the red line). The high values are not observed in the latest product version of 1.2 (Figure R2b). It should be noted that we used a substitute BPNN model in the latest GIMMS LAI4g product for winter for ENF, which has provided better LAI estimation.

[Figure]

(a)

(b)

**Figure R2.** Annual averaged LAI values from GIMMS LAI4g in the version of 1.0 (a) and 1.2 (b) in the Catchment 221201. In (a), averages were calculated from all values (black line) or values excluding the fill value of 65535 (red line).

In Figure R3, we showcase the existence of the fill value (65535) within Catchment 221201 in the first half of July 1991. Thus, we strongly recommend using the latest version of GIMMS LAI4g (v1.2) and properly processing the fill value in the analysis.

[Figure]

**Figure R3.** An example of the existent fill value (65535) within Catchment 221201 in July 1991.

**[Comment 3]** *The unusually high LAI values, averaged at the annual frequency are widespread and include intermittent lakes, see Figure 2 for 1992.*

[Figure]

Figure 2 Annual-averaged pixel LAI for southeast Australia, the colour ramp units are in m²/m².

**[Response 3]** For the abovementioned reason, unusually high LAI averages could exist whenever the algorithm could not retrieve a meaningful LAI estimate over a particular time or region (including intermittent lakes). This is not limited to the old version (v1.0) of GIMMS LAI4g but also to the latest version (v1.2). In terms of version 1.0, Figure R4

shows the annual-averaged LAI map for southeast Australia in 1992 before (a) and after (b) removing the fill value (65535). Unusually high LAI averages are marked by those > 7 $m^2/m^2$ (red), as 7 $m^2/m^2$ has been set as the LAI up limit in the GIMMS LAI4g. The black pixels beside the sea in Figure R4 mark permanent non-vegetation regions such as lakes.

[Figure]

(a)                                                                                  (b)

**Figure R4.** Annual-averaged LAI from GIMMS LAI4g (V1.0) for southeast Australia before (a) and after (b) removing the fill value (65535).

**[Comment 4]** *We further assessed this by obtaining average continental values during August (when snow cover is generally at its largest extent) of 1991 and 1992, and masking all pixels with LAI values >50, see Figure 3. Besides Alpine areas with snow cover, the unusually high values are evident in intermittent dry lakes as well.*

[Figure]

Figure 3 August-averaged pixel LAI for Australia, LAI values above 50 are shown in in yellow. Red dot in the southeast denotes a pixel used to assess the entire timeseries.

*We picked one yellow-masked pixel in the Alpine area to assess this issue in years other than 1991 and 1992. Figure 4 suggest that the issue is not present in other years.*

[Figure]

Figure 4 Monthly pixel LAI for red pixel in Figure 3, Y-axis units are in m²/m².

*The limited investigation of high LAI values in 1991 and 1992 suggest that this is limited to landscapes with high reflectance (snow, dry lake beds) and that filtering these values as for other years is a suitable solution.*

**[Response 4]** We also map the LAI variation in August 1991 (GIMMS LAI4g version 1.0) for Australia (Figure R5). Similar to Figure R4, LAI values $> 7$ $m^2/m^2$ at vegetation locations are marked by red. Those unusually high LAI values are mainly distributed in southeast Australia (Figure R5a). They could be easily removed by excluding the fill value (65535) (Figure R5b). In August 1992, similar patterns were found (not shown here).

[Figure]

(a)

[Figure]

(b)

**Figure R5.** August-averaged LAI in 1991 from GIMMS LAI4g (V1.0) for Australia before (a) and after (b) removing the fill value (65535).

The unusually high LAI values are rare in the latest version of GIMMS LAI4g (v1.2), though still exist (Figure R6a), and they could also be removed (Figure R6b).

[Figure]

(a)

[Figure]

(b)

**Figure R6.** August-averaged LAI in 1991 from GIMMS LAI4g (V1.2) for Australia before (a) and after (b) removing the fill value (65535).

Although the unusually high LAI values were found primarily in 1991 and 1992 for the case of this report, they could exist in other regions and other years if the fill value (65535) is not properly processed. We hope the community can understand our rationale for replacing invalid LAI values with the fill value (65535), which has been a common practice in remote sensing-based data products (e.g., MODIS LAI). As suggested by the research team in CSIRO, we recommend other users to properly handle the fill value as well as the quality control flag in the data analysis.

**References:**

BoM, 2022. Hydrologic Reference Stations, Bureau of Meteorology. http://www.bom.gov.au/water/hrs/, last access: July 2022.

---

## Author Response (AR4)

**Reply to Editors**

Dear Dr. Dalei Hao and Polina Shvedko,

We would like to sincerely express our gratitude for your work in processing our manuscript entitled "Spatiotemporally consistent global dataset of the GIMMS Leaf Area Index (GIMMS LAI4g) from 1982 to 2020" (essd-2023-68) submitted to *Earth System Science Data*. We have carefully considered the **file validation comments** and made modifications to the manuscript accordingly.

Below we provide point-to-point responses, each following the specific comment from the editor. We hope that the modified manuscript can meet the publication standard in ESSD.

Sincerely yours,

Zaichun Zhu, Ph. D. (on behalf of the author team)

School of Urban Planning and Design

Peking University

Tel: 86 185 0042 6608

Email: zhu.zaichun@pku.edu.cn

**[Comment 1]** *1. Your reference list includes works "in preparation" or "in review". Such works can be cited upon submission if being available to the reviewers. They should not be cited in the final, accepted manuscript, unless published, accepted for publication, or available as preprint with a DOI.*

**[Response 1]** We thank the editor for pointing this out. The work "in preparation" in the previous version of the manuscript refers to Zha et al. (in preparation):

Zha, J., Li, M., Zhu, Z., Cao, S., Zhang, Y., Zhao, W., and Chen, Y.: Spatiotemporally consistent global Landsat leaf area index validation dataset, in preparation. [see review asset]

This work has recently been posted as a preprint in EarthArXiv with a DOI and we have updated the in-text citation (Zha et al., 2023) and the reference in the revised manuscript:

Zha, J., Li, M., Zhu, Z., Cao, S., Zhang, Y., Zhao, W., and Chen, Y.: A direct evaluation of long-term global Leaf Area Index (LAI) products using massive high-quality LAI validation samples derived from Landsat archive, EarthArXiv [preprint], https://doi.org/10.31223/X58T05, 13 September 2023.

**[Comment 2]** *2. Please ensure that the colour schemes used in your maps and charts allow readers with colour vision deficiencies to correctly interpret your findings. Please check your figures using the Coblis – Color Blindness Simulator (https://www.color-blindness.com/coblis-color-blindness-simulator/) and revise the colour schemes accordingly.*

**[Response 2]** Thanks for this message. We have used the Coblis – Color Blindness Simulator to double-check all the figures in the manuscript and supplementary materials. Color schemes in Figure 7, Figure 11, Figure S15, and Figure S16 have been updated accordingly, as below:

[Figure]

**Figure 7.** Inter-comparison of spatially averaged LAI along latitude between the GIMMS LAI4g, GIMMS LAI3g, GLASS LAI, and GLOBMAP LAI in January and July of the years 1990, 2000, and 2010. The spatial average was calculated at an interval of 1°.

[Figure]

**Figure 11.** Annual anomalies and trends of GIMMS LAI4g before consolidation (1982−2015), GIMMS LAI4g after consolidation (1982−2020), Reprocessed MODIS LAI (2004−2020), and PKU GIMMS NDVI (1982−2015). Note that the regression equations

within the square brackets were calculated from different periods depending on the products.

[Figure]

**Figure S15.** Annual anomalies ($m^2m^{-2}$) and trends of GIMMS LAI4g before consolidation (1982−2015), GIMMS LAI4g after consolidation (1982−2020), Reprocessed MODIS LAI (2004−2020), and PKU GIMMS NDVI (1982−2015) for different vegetation biome types.

[Figure]

**Figure S16.** Annual anomalies ($m^2 m^{-2}$) and trends of GIMMS LAI4g before consolidation (1982−2015), GIMMS LAI4g after consolidation (1982−2020), GIMMS LAI3g (1982−2015), GLASS LAI (1982−2015), and GLOBMAP LAI (1982−2015) in the Congo (a) and Amazon (b) forests.

**[Comment 3]** *3. It seems that table is included as figure #10. If it is so, it must be re-labelled as table and the references in the manuscript text must be adjusted accordingly. A table may be inserted as an image, but still be called as a table.*

**[Response 3]** We apologize for the confusion. In Figure 10, we intended to use colors to better interpret the results (LAI accuracy differences between periods). However, the colors may not be well presented and the figure was more like a table. In the revised manuscript, Figure 10 has been improved as below.

[Figure]

**Figure 10.** Temporal consistencies between different periods for the global LAI products. The global LAI products include GIMMS LAI4g, GIMMS LAI3g, GLASS LAI, and GLOBMAP LAI). The periods are 1984−2015 (p1), 1984−2000 (p2), and 2001−2015 (p3). The consistencies were evaluated at the biome level using $R^2$ (a), RMSE (b), MAE (c), and MAPE (d) calculated based on Landsat LAI samples. GLO represents the global vegetation biome.